

# Continuous vegetation record of the Greater Cape Floristic Region (South Africa) covering the past 300 thousand years (IODP U1479)

Lydie M. Dupont[1,7], Xueqin Zhao[1,8], Christopher Charles[2], J. Tyler Faith[3,4], David R. Braun[5,6]

[1] MARUM - Center for Marine Environmental Sciences, University of Bremen. Leobener Str. 8, 28359 Bremen, Germany
[2] The Scripps Institution of Oceanography, University of California San Diego, La Jolla, CA 92037, USA
[3] Natural History Museum of Utah, University of Utah, Salt Lake City, UT 84108, USA
[4] Department of Anthropology, University of Utah, Salt Lake City, UT 84112, USA
[5] Center for the Advanced Study of Human Paleobiology, George Washington University, Washington DC 20052, USA
[6] Department of Archaeology, University of Cape Town, Rondebosch 7701, South Africa
[7] present address: Hendrik Werkmanstr. 5, 1061 VA Amsterdam, Netherlands
[8] present address: College of Marine Sciences, Shanghai Ocean University, Shanghai, China

*Correspondence to*: Lydie M. Dupont (dupont@uni-bremen.de)

**Abstract.** The flora of the Greater Cape Floristic Region (GCFR) of South Africa is a biodiversity hotspot of global
significance, and its archaeological record has contributed substantially to the understanding of modern human origins. For
both reasons, the climate and vegetation history of south-western South Africa is of interest to numerous fields. Currently
known paleo-environmental records cover the Holocene, the last glacial-interglacial transition and parts of the last glaciation
but do not encompass a full glacial-interglacial cycle. To obtain a continuous vegetation record of the last Pleistocene glacial-
interglacial cycles, we studied pollen, spores and micro-charcoal of deep-sea sediments from IODP Site U1479 retrieved from
SW of Cape Town. We compare our palynological results of the Pleistocene with previously published results of Pliocene
material from the same site. We find that the vegetation of the GCFR, in particular Fynbos and Afrotemperate forest, respond
to precessional forcing of climate. The micro-charcoal record confirms the importance of fires in the Fynbos vegetation.
Ericaceae-rich and Asteraceae-rich types of Fynbos could extend on the western part of the Palaeo-Agulhas Plain (PAP), which
emerged during periods of low sea-level of the Pleistocene.

## 1.    Introduction

Southwestern South Africa is a fascinating region for both its unique flora and for its role in our knowledge of the origin of
modern human behaviour (Figure 1). The Greater Cape Floristic Region (GCFR) harbours an exceptionally rich flora with
many endemics, which has been relatively stable since the Late Miocene (Linder, 2003; Dupont et al., 2011). The region also
has a very rich archaeological record (e.g., Henshilwood et al., 2002; 2011; Mackay et al., 2014; Marean et al., 2014), especially
of the last climate cycle (from 150 thousand years ago). The paleo-environmental records of the GCFR sites covering at least
part of the last glaciation are varied and detailed. They include stable carbon and oxygen isotope records of speleothems (Talma
and Vogel, 1992; Bar-Matthews et al., 2010; Braun et al., 2019; 2020), stable isotopes from hyrax middens (Chase et al., 2011;





2012; 2017; 2018; 2019), mammal bone assemblages (Klein, 1983; Avery, 1982; Faith, 2013; Klein and Cruz-Uribe, 2016, 2000; Nel and Henshilwood, 2016; Nel et al., 2018; Forrest et al., 2018), microwear patterns and stable isotopes of fossil teeth

(Copeland et al., 2016; Sealy et al., 2016; 2020; Hodgkins et al., 2020; Williams et al., 2020), charcoal (Cartwright and Parkington, 1997; Cowling et al., 1999; Cartwright, 2013; Parkington et al., 2000), phytoliths (Esteban et al., 2018; 2020) and pollen (Meadows and Sudgen, 1991; Chase and Meadows, 2007; Scott and Woodborne, 2007a; 2007b; Meadows et al., 2010; Quick et al., 2011; 2015; 2016; Valsecchi et al., 2013; Chase and Quick, 2018; Scott and Neumann, 2018). The list is by no means comprehensive.

After considerable efforts from various disciplines, detailed maps of the now submerged continental shelf area south of South Africa have been constructed (Cleghorn et al., 2020) providing a view of the ecosystems on the Palaeo-Agulhas Plain (PAP), which was largely exposed during the last glacial maximum (Marean et al., 2020). The inferred glacial vegetation of the PAP is based on vegetation modelling (Cowling et al., 2020) using the geological and soil maps (Cawthra et al., 2020) in connection with regional climate modelling (Engelbrecht et al., 2019). A wide landscape emerged divided by broad river systems with

diverse Fynbos types in the western and southern part of the PAP. The central portion of the PAP was probably dominated by open grasslands.

Despite the wealth of information, a synthetic understanding of vegetation and climate of the past glacial-interglacial cycle remains elusive. This uncertainty stems from a combination of problems including (i) uneven spatiotemporal sampling coupled with a substantial amount of poorly understood spatial variation (e.g., Chase et al.,2017; 2018); (ii) records that reflect changes

over limited spatial extend and over relative short periods of time (e.g., Chase, 2010; Marean et al., 2014); (iii) reliance on a diversity of different proxies, not all of which directly inform about the climate or vegetation parameters of interest (Chase et al., 2013); (iv) seemingly contradictory lines of evidence (e.g., Avery 1982; Faith, 2013) as well as conflicting interpretations of the evidence (e.g., Faith et al., 2019; Thackeray, 2020).

To provide a regional framework of climate and vegetation dynamics across multiple glacial-interglacial cycles, we report here

a record of the vegetation in the GCFR using pollen and spores recovered from deep-sea sediments of IODP Site U1479 (Figure 1), which were retrieved offshore almost 160 km SW of Cape Town (Hall et al., 2017). While the terrestrial paleontological and palaeobotanical record mostly reflect the environment of a relatively limited spatial area and cover relative short stretches in time (Marean et al., 2014) marine sediments collect pollen and spores from a wide terrestrial area and thus record an integrated regional signal (Dupont, 2011). Moreover, the marine sediments have a chronology that allows the assessment of

vegetation dynamics with respect to orbital forcing and other climatic proxy records that are tied to the same chronology. The sediments of IODP Site U1479 demonstrably provide a continuous signal extending from the late Quaternary down to the Pliocene. Our record enables us to continuously follow regional vegetation change in the GCFR over the past 300 thousand years (ka) at a temporal resolution of ca 3 ka. We investigate the vegetation response to climate changes and sea-level fluctuations of the middle to late Pleistocene climate cycles.






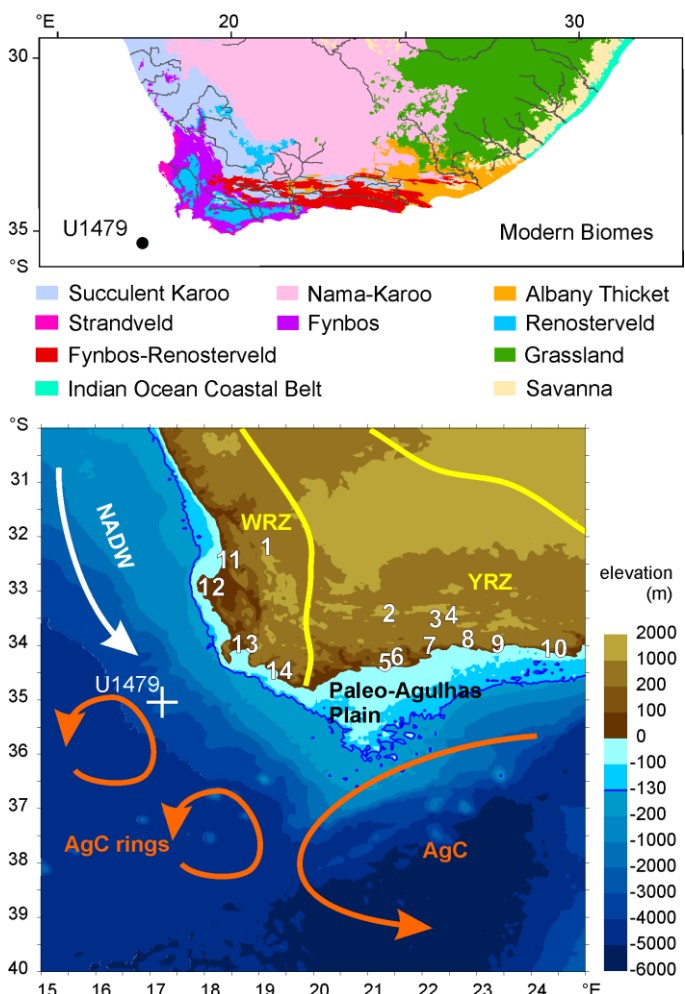

**Figure 1. Top. Modern vegetation of South Africa south of 30°S with main rivers draining to the ocean (Mucina and Rutherford, 2006). Bottom. Map of the southern and western Cape and sites mentioned in the text. The direction of the North Atlantic Deep Water (NADW), the Agulhas Current (AgC), the Agulhas rings (AgC rings) and Palaeo-Agulhas Plain (PAP) are iIndicated over the ocean; on the continent the winter rainfall zone (WRZ) and the year-round rainfall zone (YRZ). The location of IODP Site U1479 is denoted by the white cross. 1. Cederberg Mountains, 2. Seweeekspoort, 3. Boomplaas and Cango Caves, 4. Efflux Cave, 5. Blombos Cave, 6. Still Bay (Rietvlei), 7. Pinnacle Point, 8. Wilderness area (Vankervelsvlei), 9. Nelson Bay Cave, 10. Klasies River Mouth Cave, 11. Elands Bay Cave, 12. Hoedjiespunt, 13. Swartklip, 14. Die Kelders Cave**.

## 1.1 Modern climate and vegetation

The modern climate of South Africa is dominated by the atmospheric South Atlantic High-pressure system in the southwest and the easterly flow from the Indian Ocean (Tyson and Preston-Whyte, 2000). This results in a climate pattern over South Africa that can be divided according to its rainfall regime (Figure 2) (Hijmans et al., 2005; Chase and Meadows, 2007). In most of South Africa, precipitation falls during the summer months (October to March; Summer Rainfall Zone, SRZ), while in the south-western part of the sub-continent, westerly winds bring precipitation during the winter months (April to September;





Winter Rainfall Zone, WRZ). A transitional region located between these two regions receives rain in all seasons (Year-round Rainfall Zone, YRZ). Furthermore, sea surface temperatures influence the climate, in particular the warm waters of the Agulhas Current and the cold waters of the Benguela upwelling system.

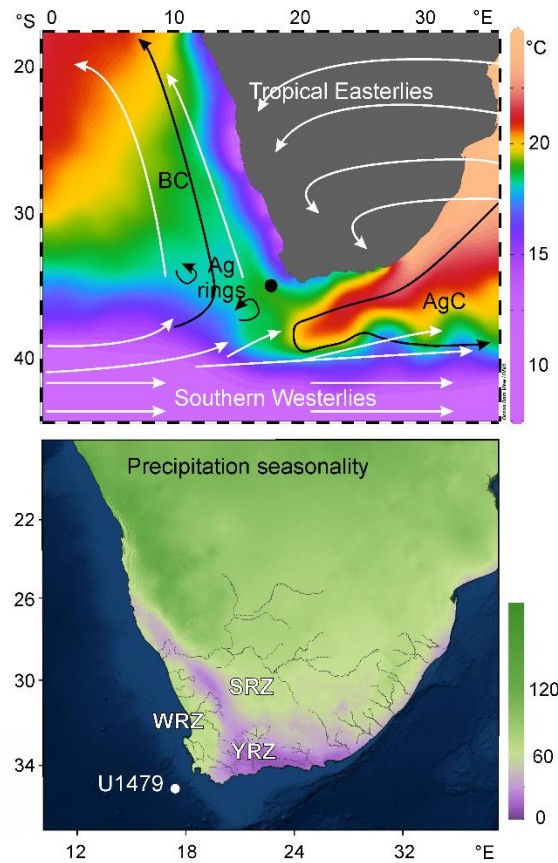


**Figure 2. Top. Map of modern atmosphere and surface ocean circulations with modern sea surface temperatures (World Ocean Atlas 2013, Boyer et al., 2013) and the location of IODP Site U1479 (dot). BC, Benguela Current; AgC, Agulhas Current; Ag rings, Agulhas rings. Bottom. Precipitation seasonality (mean monthly precipitation coefficient of variation) showing three rainfall zones in southern Africa derived from WorldClim version 1.3 (Hijmans et al., 2005). WRZ, winter rainfall zone; SRZ, summer rainfall**
**zone; YRZ, year-round rainfall zone (lilac).**

While modern rainfall in the SRZ is modulated by the sea surface temperature (SST) of the Indian Ocean (Reason and Mulenga, 1999), south-western African precipitation is affected by South Atlantic SST and the Agulhas Current. Winter rainfall in the south-western Cape comes mainly with cold fronts, especially if the westerlies have a northern position over the southern tip
of Africa. Warm Atlantic SSTs provide more moisture to be advected into the continent. Moisture from the warm Agulhas waters recirculates over southernmost Africa (Reason et al., 2003).

The situation on the south coast is more complicated. Both tropical-temperate troughs from the east and frontal systems from the west influence the balance between WRZ and YRZ. According to Engelbrecht et al. (2015), modern precipitation along





the south coast is contributed by cut-off-lows (16% of annual rainfall), ridging high pressure systems (46%) and tropical-
temperate troughs (28%). In addition, the Agulhas Current would have influenced the coastal area propagating climate signals
from the tropics to the southern Cape coast (Chase and Quick, 2018).

The GCFR is situated in the WRZ and YRZ and encompasses three biomes, Fynbos (including Fynbos *sensu stricto*,
Renosterveld and Strandveld), Succulent Karoo and Albany thicket (Figure 1). The semi-desert vegetation of the Nama Karoo
grows in the SRZ northeast of the GCFR (Mucina and Rutherford, 2006). Fynbos consists of fine-leaved (from the Afrikaans
word meaning "fine bush") evergreen shrubs growing on nutrient poor soils of the south-western Cape with annual
precipitation between 600 and 800 mm. It is extraordinarily rich in Erica species. In addition to Ericaceae other important
families are Asteraceae, Rhamnaceae, Thymelaeaceae, Rutaceae, Proteaceae and the almost endemic Bruniaceae. The most
important graminoids are the Restionaceae (Cape reeds) but Cyperaceae (sedges) and Poaceae (grasses) also occur.
Renosterveld is dominated by evergreen asteraceous shrubs, notably *Elytropappus rhinocerotis* (renosterbos). *Cliffortia*
(Rosaceae) and *Anthospermum* (Rubiaceae) are important elements and so are species of Fabaceae and Malvaceae. Along the
coast the littoral thicket, called Strandveld, shrubs and small trees are present that have broader leaves than the shrubs of the
Fynbos. In the thicket, apart from the trees (Celastraceae, *Diospyros*, *Morella*) Restionaceae are abundant (Rebelo et al., 2006).
The Succulent Karoo vegetation contains many dwarf leaf-succulents of Aizoaceae and Crassulaceae growing in the drier parts
of the GCFR, in which the sparse rain (average 170 mm/a) falls predictably during the winter months. Furthermore, the
vegetation includes Asteraceae, Amaranthaceae, Euphorbiaceae and Zygophyllaceae but few grasses (Mucina et al., 2006).
Albany Thicket grows in the eastern part of the GCFR and is probably less relevant for this study.

On the south and east facing slopes of the Cape Fold Belt Mountains and along the south coast Afrotemperate forest is growing,
which has a multi-layered canopy and occurs in the deep gorges. Podocarpaceae species (*Podocarpus latifolius* and *Afrocarpus
falcatus*) are important constituents. Afrotemperate forest needs mean annual rainfall of at least 525 mm (Mucina and
Geldenhuys, 2006). In both Fynbos and Renosterveld, burning of the vegetation is important and its regeneration depends on
fire. Fires are less important in the Succulent Karoo and Afrotemperate forest due to insufficient fuel in the former and too
much moisture in the latter (Whitlock et al., 2010).

## 2    Material and Methods

IODP Site U1479 (35°04′S, 17°24′E) is located ca 160 km southwest of Cape Town, South Africa, on a 30 km wide ridge
rising ca 200 m above its surroundings on the mid-to-lower western slope of the Agulhas Bank in the Cape Basin at a water
depth of ca 2615 m below sea level (Figure 1, Hall et al., 2017). The seafloor at the site is bathed in North Atlantic Deep Water,
while the surface ocean is influenced by the waters of the Agulhas Current, which retroflects from the Agulhas Bank at the
southern tip of Africa and then flows eastwards as the Agulhas Return Current. The Agulhas retroflection occurs east of Site
U1479 (Hall et al., 2017), but the phenomenon is characterized by the frequent shedding of warm core rings that pass directly





over the site. A principal motivation for drilling Site U1479 was to develop a long record of this Agulhas "leakage" of warm

and salty Indian Ocean water into the South Atlantic Ocean.

Nine holes were cored at Site U1479 to ensure a continuous sedimentary sequence spanning more than 230 meters (early

Pliocene to recent). A spliced composite record from these nine holes was created from multiple physical properties measured

shipboard, most notably sediment colour (Hall et al., 2017). Sampling for various micro-paleontological and geochemical

purposes was conducted along this spliced composite. The chronostratigraphic framework for this splice derives from the

oxygen isotopic composition of the benthic foraminifera *Planulina wuellerstorfi*, measured at 5 cm intervals at the Scripps

Institution of Oceanography using a Thermo MAT253 equipped with a Kiel IV carbonate preparation device. Analytical

uncertainty is less than 0.08‰. The details of the full record (including the accompanying carbon isotopic analyses) are

presented elsewhere (Hines et al., 2021), but the interval corresponding to the last 300 ka is displayed in Figures 3 and 4.


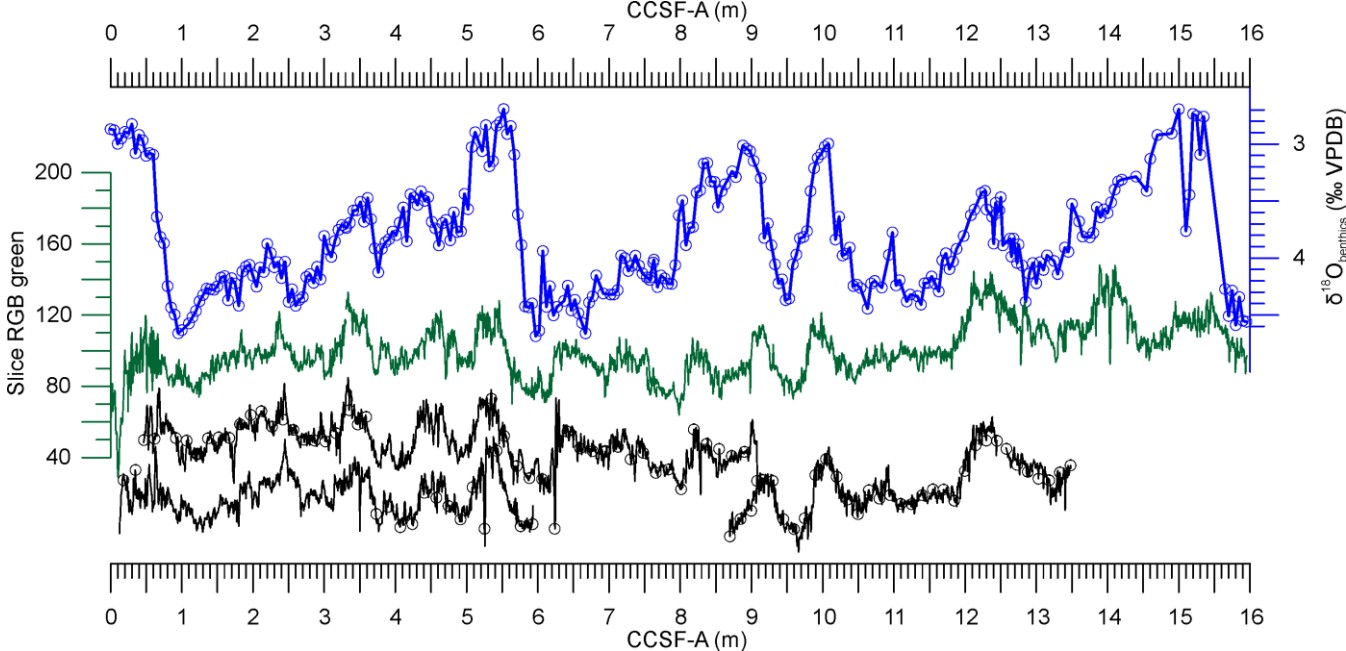

**Figure 3. Correlation of upper cores of Holes C, B, G, H and I of IODP Site U1479 (black) with the splice (green) using the green channel of the colour scan (Hall et al., 2017 and Shipboard data). Black curves are offset vertically for clarity. Positions of the pollen samples are denoted with black circles. On top the stable oxygen isotopes of *Planulina wuellerstorfi* in blue. VPDB, Vienna standard**

**PeeDee Belemnite. For more detail see Supplementary Figure 1.**

The core material from the shipboard splice was largely exhausted by this initial sampling. However, in the upper part of the

sequence, there was sufficient overlap among the cores recovered to construct a parallel spliced composite record. Accordingly,

we selected samples from Holes B, C, G and H to obtain a record of the past 300 ka with a time step of 2 to 4 ka. Holes were

correlated using the green channel values of the shipboard data (Figure 3, Supplementary Figure 1). We first plotted the green

values of Cores B1, C2, G1, H1, H2 and I1 against the published composite depth (splice). Because the hole compositing



approach involves single depth offsets for an entire core (ca 10 meter in length), it does not account for variable stretching/compression of cored intervals relative to their stratigraphic correlates in a different spliced composite. In order to make use of the stratigraphic information of oxygen isotopes measured in the shipboard splice, it is necessary to account for

this relative distortion in our alternative splice. The depth of Core B1 had to be shifted 46 cm upwards and stretched by a factor of 1.184 to obtain the best stratigraphic correlation with the shipboard splice. Depths of Core H1 only had to be stretched by a factor of 1.127 to achieve optimal stratigraphic correlation Supplementary Figure 1).

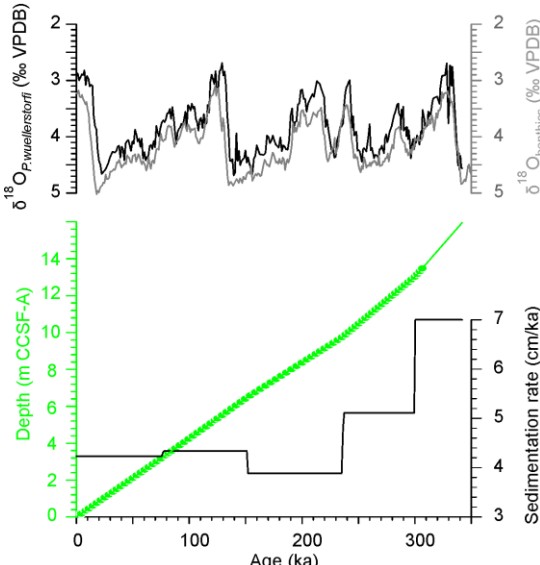

**Figure 4. Age depth model of the upper 14 m of IODP Site U1479 based on tuning of colour channel data and resulting sedimentation rates. Green dots denote pollen samples. On top the stable oxygen isotopes of benthic foraminifera of Site U1479 (black) and global stack LR04 (grey, Lisiecki and Raymo, 2005). VPDB, Vienna standard PeeDee Belemnite.**

The chronology for the spliced composite can be established in a number of ways, including transferal of common oxygen

isotopic chronologies (e.g., LR04 stack, Lisiecki and Raymo, 2005). However, the chronology we employ here uses a more independent approach. Throughout the ca 6 Ma long record of Site U1479, sediment colour is demonstrably coherent with climatic precession, and, therefore, the chronology can be tuned to maximize the coherence of colour channel data (e.g., RGB green) with climatic precession; the phase of the color with respect to orbital variability is fixed by radiocarbon constraints of the last 30 ka. Discrete shipboard measurements and XRF scanning indicate that the sediment colour essentially monitors

variable carbonate content, but the orbital tuning approach requires no assumption about the actual mechanism through which orbital variability paces the carbonate variability. The resulting age-depth model and sedimentation rates for the past 350 ka are plotted in Figure 4, illustrating that sedimentation rates implied by orbital tuning are relatively stable around a mean of ca 4.5 cm ka$^{-1}$ since ca 300 ka. Furthermore, we can use the oxygen isotopic record to cross check the orbitally tuned chronology against the LR04 global isotope chronology (Lisiecki and Raymo, 2005); the differences are generally within error (Figure 4)



though there is isolated divergence in discrete intervals such as the penultimate deglaciation. At the top of the sequence, the later Holocene is probably missing. In any case, through either approach—oxygen isotopic or orbital tuning—the sediment chronology clearly resolves the suborbital variability of the last three ice age cycles.

Sample volume was measured using water displacement. Pollen samples were decalcified with cold HCl (ca 10%) and then treated with HF (ca 40%) for two days. Concentrated HCl (ca 37%) was added to keep fluor-complexes in solution. Samples
were sieved over a 7-µm nylon screen while ultrasonically disaggregating organic matter. The residuals were stored in water, mounted in glycerol and examined under a light microscope (magnification 400 x and 1000 x) for pollen, spores and micro-charcoal. Two *Lycopodium* spore tablets (batch No. 177745 containing 18584 ± 829 spores per tablet) were added during the decalcification step.

Pollen grains were identified using the African pollen reference collection of the Department of Palynology and Climate
Dynamics of the University of Göttingen, the photo collection of the Department of Environmental and Geographical Science of the University of Cape Town, the African Pollen Database (http://apd.sedoo.fr/pollen/interface/indexPollen.html) and literature (Bonnefille and Riollet, 1980; Scott, 1982; Schüler and Hemp, 2016). Black opaque angular particles were counted as micro-charcoal.

Percentages were calculated based on the sum of pollen and spores (ranging between 141 and 360). Pollen concentration was
determined based on the Lycopodium spore counts and accumulation rates by multiplying the concentration with the sedimentation rate. The 95% confidence intervals of percentages were calculated following Maher (1972), those of concentration values after Maher (1981). Pollen zones were determined using stratigraphically constrained cluster analysis (unweighted pair-group average and correlation similarity index) from the PAST package (Hammer et al., 2001). The latitudinal insolation gradient (LIG) during Southern Hemisphere (SH) summer and winter was calculated using the insolation
data of La04 (Laskar et al., 2004); SH summer LIG by subtracting the insolation at 30°S from that at 60°S on December 21 (270°) and SH winter LIG by subtracting the insolation at 30°S from that at 60°S on June 21 (90°). To compare pollen concentrations with curves of possible forcing mechanisms, such as SH summer LIG, SH winter LIG, stable oxygen isotopes and sea-level, linear correlations were calculated on the equidistantly interpolated values resampled every 3 ka. Also using PAST, spectral analysis was carried out with REDFIT (Schulz and Mudelsee, 2002), which does not need equidistantly spaced
data. We employed a Hanning window, 2 times oversampling and 2 segments resulting in a bandwidth of 0.00993 ka$^{-1}$. To determine phase shifts between pollen data and insolation curves, we employed Blackman-Tukey cross-spectral analysis using ARAND software (Howell et al., 2006). Data were interpolated equidistantly and resampled every 3 ka between 3 and 306 ka. We applied 40 lags resulting in a bandwidth of 0.01111 ka$^{-1}$.

Introduction





## 205  3    Palynological results

The main pollen types found are Restionaceae (7-27%), Ericaceae (1-11%) and *Stoebe-Elytropappus* type (2-11%). Other taxa from Fynbos include Proteaceae, Rhamnaceae, Bruniaceae, *Passerina*, and *Cliffortia*. Drought-adapted taxa are less abundant but consistently present, and include Aizoaceae (up to 5%), *Ruschia* type, Crassulaceae, *Pentzia-Cotula* type (up to 6%), Amaranthaceae (up to 8%), *Tribulus* and *Zaluzianskya* type (compare *Selago* type of Scott 1982). The most abundant type in

some parts is Podocarpaceae pollen (up to 46%), with other Afrotemperate forest elements including pollen of *Olea*, *Myrsine* and Meliaceae/Sapotaceae. Fern spores from Anemiaceae and *Cyathea* type might also have their source in the forest. Elements from coastal thicket include *Rhus* type, *Buxus*, *Celtis*, Combretaceae, Ebenaceae, Euphorbiaceae pp, *Kiggelaria-Spirostachys* type, *Antidesma*, *Morella* and *Dodonaea*. Other important pollen types are Asteraceae pp (6-22%), Cyperaceae (8-27%), Poaceae (1-10%) and *Anthospermum* (up to 5%). The percentage pollen diagram is divided into nine zones based on

stratigraphically constrained cluster analysis (Figure 5, Table 1, Supplementary Figure 2).

**Table 1. Pollen zone description. (* MIS, marine isotope stage; T, termination)**

| Zone | N | Age (ka) | marine stratigraphy* | Description based on pollen percentages |
|---|---|---|---|---|
| **VII** | 13 | 307-274 | late MIS9 | Podocarpaceae decline from 29% to 10% but have a maximum at the end of the zone, *Olea* and *Morella* increase, Asteraceae pp increase up to 16%, *Anthospermum* increase up to 3%, Aizoaceae increase slightly (2%), Ericaceae maximum (9%) in the middle of the zone |
| **VI** | 11 | 271-242 | MIS8, T-IIIb, early MIS7e | Podocarpaceae and thicket taxa are low with exception of *Morella*, Asteraceae pp have a maximum (20%) in the beginning of the zone, Ericaceae maxima up to 11%, *Passerina* maximum (3%), increasing *Anthospermum* up to 4%, Aizoaceae increase (3%), increasing maxima of *Pentzia-Cotula* type and Amaranthaceae up to 6 and 8% respectively, *Zaluzianskya* type maximum (2%) |
| **V** | 4 | 239-228 | end of MIS7e, MIS7d | Podocarpaceae maximum (36%), small maximum of *Olea* (2%), decline of *Anthospermum* to 1%, *Pentzia-Cotula* type decline to 2% |
| **IVb** | 23 | 224-142 | T-IIIa, MIS7a-c, most of MIS6 | Podocarpaceae vary between 3-22%, other forest and thicket taxa occur mainly in the early part of the zone with exception of *Morella*, Restionaceae maximum of 24% at the beginning, Asteraceae pp increase up to 22%, Ericaceae relatively high (mean 6%), *Cliffortia* increases and *Passerina* reaches over 2%, *Stoebe-Elytropappus* type increase to maximum of 11%, Proteaceae decline, Amaranthaceae maxima up to 6% early in the zone, Cyperaceae maximum (27%) |
| **IVa** | 3 | 139-135 | end of MIS6 | Podocarpaceae maximum (26%), *Anthospermum* maximum (5%), short *Zaluzianskya* type maximum, decline of Ericaceae |
| **IIIb** | 4 | 133-125 | T-II, early MIS5e | Podocarpaceae minimum of 2%, *Kiggelaria-Spyrostachys* type maximum (1%), most thicket elements occur, Poaceae maximum of 9%, Restionaceae increase up to 22%, Aizoaceae pp. maximum (5%), Amaranthaceae maximum (4%), *Euphorbia* maximum (2%) |
| **IIIa** | 17 | 123-64 | MIS5 from the middle of MIS5e, early MIS4 | Podocarpaceae maxima >40% (mean 29%), other forest and thicket elements present but low, Restionaceae vary between 7-20%, most Fynbos and drought-adapted taxa low, *Pentzia-Cotula* type and Amaranthaceae decline from 3% at the beginning to 0 and 1% at 88 ka respectively, Olea maximum at the end of the zone |
| **II** | 10 | 61-29 | late MIS4, MIS3, | Asteraceae pp increase up to 21%, Ericaceae increase (3-9%), *Passerina* increase up to 2%, *Stoebe-Elytropappus* type slightly increase (5-9%) |
| **I** | 6 | 25-4 | MIS2, T-I, MIS1 | Podocarpaceae minimum (2%), Restionaceae maximum (27%), early in the zone *Pentzia-Cotula* type (6%) and *Anthospermum* (4%) maxima, Amaranthaceae (6%) and Aizoaceae pp (5%) maxima late in the zone, forest elements such as Combretaceae and *Kiggelaria-Spyrostachys* type increase |





Total pollen concentration varies between 147 and 1168 grains/ml (Figure 5). Concentration of most pollen types remains
below 100 grains/ml but for Restionaceae (26-125 grains/ml), Cyperaceae (27-235 grains/ml), Podocarpaceae (6-485
grains/ml) and Asteraceae pp (11-130 grains/ml). With the exception of the Asteraceae, these are wind-pollinated plants.
However, concentrations of wind-pollinated Poaceae pollen are relatively low (8-77 grains/ml). Charcoal particle
concentration varies between 2600 and 18000 particles/ml (Figure 5). Long micro-charcoal particles make up only a fraction
of the total with ratios fluctuating between 3 and 18% but for two samples dated 3 and 7 ka, in which 21% and 25% of the
micro-charcoal particles are longish. Average sedimentation rates decrease from 7 to 5 cm/ka around 300 ka and decline to ca
4 cm/ka after 240 ka (Figure 4).

We performed spectral analysis on the logarithmic transformed accumulation rates [log(ARs)] to avoid interdependence
between the data as is the case with percentages. We analysed the accumulation rates of total pollen and spores, micro-charcoal,
Restionaceae, Asteraceae pp, Poaceae, Cyperaceae, Ericaceae, Podocarpaceae, *Stoebe-Elytropappus* type, *Anthospermum*,
*Pentzia-Cotula* type and fern spores (Tables 2 and 3). Accumulation rates are given in Supplementary Figure 3, and significant
power maxima in Table 2. The spectral density (power) of the accumulation rates of these taxa exceeds the 90% $X^2$ significance
level in the precession band. Additional significant power in the obliquity band is found for Restionaceae, Poaceae and *Stoebe-
Elytropappus* type. Cross-spectral analysis revealed these taxa to be negatively correlated with the Southern Hemisphere winter
latitudinal insolation gradient (SH winter LIG) lagging between 0 and 3 ka (0-42° phase shift). Highest coherency was found
for the cross-spectrum of Podocarpaceae pollen accumulation rates with SH winter LIG multiplied by minus 1 (Table 3).

## 4    Discussion

### 4.1    Source area and pollen transport

The floral composition of the palynological assemblage of Site U1479 indicates that it records the biomes of the GCFR of
South Africa (Figure 5, Supplementary Figure 2). The Fynbos biome is represented by pollen from Restionaceae, Ericaceae,
Proteaceae, Rhamnaceae, Bruniaceae, Passerina and Cliffortia. Pollen of *Stoebe-Elytropappus* type (here probably from
*Elytropappus*, the renosterbos), *Anthospermum*, and Asteraceae pp. may also be associated with the GCFR. Drought-adapted
plants from families such as Aizoaceae, Crassulaceae, Scrophulariaceae (*Zaluzianskya* type), Euphorbia and certain Asteraceae
(*Pentzia-Cotula* type) can be found in the Succulent Karoo and the Nama Karoo. Afrotemperate forest is represented by
Podocarpaceae, *Olea* and *Myrsine*, and thicket is indicated by the presence of pollen of *Olea*, *Dodonaea*, *Morella*, *Kiggelaria-
Spirostachys* type (here probably from *Kiggelaria*), Ebenaceae, Combretaceae, *Celtis* and *Rhus*. Along the south coast, the
same pollen taxa have been found at Still Bay (Quick et al., 2015) and/or at Vankervelsvlei (Quick et al., 2016). With exception
of Podocarpaceae, the floristic composition also resembles records from the Cederberg Mountains near the border between
mountain Fynbos and Succulent Karoo (Scott and Woodborne, 2007a; 2007b; Meadows et al., 2010; Quick et al., 2011).





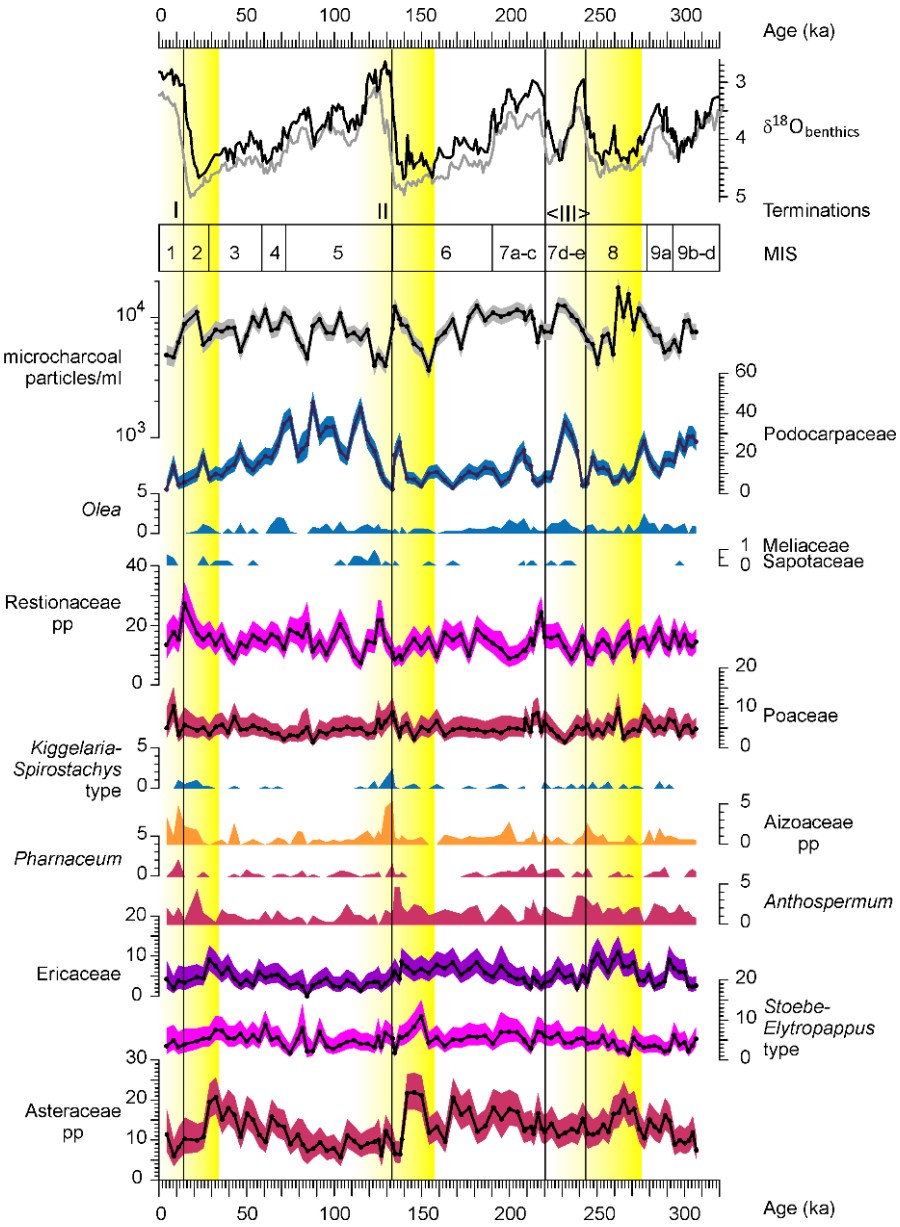


**Figure 5. Percentages of selected pollen taxa with 95% confidence intervals and micro-charcoal particle concentration per ml. On top the stable oxygen isotopes of benthic foraminifera in per mil VPDB (Vienna PeeDee Belemnite) of Site U1479 (black) and global stack LR04 (grey, Lisiecki and Raymo, 2005) and marine isotope stages (MIS). Vertical black lines denote Termination I (beginning of MIS1), Termination II (beginning of MIS 5) and the double Termination III (MIS 7). Yellow colour gradients denote the sequence**
**of taxa described in the text.**

The sediments of marine Site U1479 are thus rich in pollen and spores from southernmost Africa. This is remarkable because at present no big rivers discharge in the vicinity of the core location and the general wind direction of the southern westerlies is landwards. Moreover, the site lies outside the influence of the SE trade winds. However, numerous smaller rivers drain the





south coast of South Africa, which bring pollen and spores to the ocean. Further westward transport could derive from the
Agulhas Current, which also transports clays into the Cape Basin (Petschick et al., 1996).

This situation was different during sea-level low-stands, especially during the LGM, when a wide network of rivers spread
over the PAP (Cawthra et al., 2020). The Breede River, now discharging into the ocean near Cape Infanta west of Blombos
Cave on the south coast, watered the western part of the PAP. During the LGM, the mouth of the Breede River was probably

situated almost directly south of Cape Agulhas, some 250 km east of Site U1479 (Cawthra et al., 2020). Thus, during sea-level
low-stands the dominant source area of pollen and spores might have shifted from the present south coast and adjacent inland
to the western part of the PAP.

## 4.2 Vegetation development

For the past 300 ka, three Terminations (i.e., rapid onset of interglacial conditions) have been recognized (Cheng et al., 2009);
T-I at the beginning of MIS 1 (17-11 ka), T-II at the beginning of MIS 5 (ca 130 ka) and T-IIIb at the beginning of MIS 7 (ca
244 ka). Additionally, another Termination (T-IIIa) is placed at the beginning of MIS 7c (ca 218 ka) as implied by the
recognition of MIS 7c as a full interglacial (PAGES, 2016). Our data indicate a change in vegetation from ericaceous-rich
Fynbos with abundant Asteraceae during the glacial stages MIS 8, MIS 6, and MIS 2-4, via subsequently increased percentages
of *Anthospermum*, Aizoaceae and *Kiggelaria-Spirostachys* type pollen during the Terminations to more forest with Olea and
Podocarpaceae during the interstadial stages MIS 7d, MIS 5d-a (Figure 5). The representation of forest remained low during
the Holocene.

During the last glacial-interglacial transition, a shift from ericaceous Fynbos to a vegetation with more thicket species (*Olea*,
*Rhus*, *Euclea*, *Dodonaea*, *Myrsine*) was found in records from the Cederberg Mountains (Scott and Woodborne, 2007a; 2007b;
Quick et al., 2011; Valsecchi et al., 2013). Around the Pakhuis Pass Site in the Cederberg Mountains the vegetation changed
from abundant Proteaceae, Ericaceae, *Stoebe-Elytropappus* type, *Passerina* and Scrophulariaceae to increased Aizoaceae,
*Anthospermum*, *Dodonaea*, *Rhus*, *Euclea* (Ebenaceae), *and* Olea (Scott and Woodborne, 2007b). Although the details differ,
the marine sediments show a similar pollen sequence and might have recorded shifts in the regional abundance of Fynbos,
succulent vegetation, thicket and forest. However, it should be kept in mind that the marine sediments collect pollen and spores
from a large area and that the subsequent maxima in the marine record would not represent a local vegetation succession.

Afrotemperate forest and littoral thicket elements are present in the pollen record of Vankervelsvlei in the Wilderness area
during the period from MIS 5c to MIS 3 (Quick et al., 2016), although percentages of Podocarpaceae pollen already decline
around 80 ka. This record indicates a shift from ericaceous Fynbos and Renosterveld vegetation during the glacial period to
asteraceous Fynbos with few ticket and forest elements during the Holocene. The record of Rietvlei in the Still Bay area further
to the west (Quick et al., 2015) exhibits high representation of Restionaceae but few other Fynbos elements, limited trees and
shrubs, and some succulents (Aizoaceae) during MIS 3. The Rietvlei record continues after 16 ka with a reduced representation
of Fynbos and an increase of drought-resistant Amaranthaceae. Fynbos and Proteaceae increase at the beginning of the
Holocene (11 ka).



The subsequent changes in the vegetation around the Terminations indicate influence of the glacial-interglacial cycle. Asteraceae, Ericaceae, *Passerina* and *Anthospermum* tend to be more abundant during glacial stages. Cooler temperatures
during glacial stages might have favoured Ericaceae (Quick et al., 2016). Podocarpaceae also seem to increase during cooler stages with possibly greater moisture availability, although not during glacial maxima. Low atmospheric $CO_2$ during MIS 8, 6 and 2 might have hampered the growth of Afrotemperate forests (Bereiter et al., 2015; Dupont et al., 2019). Probably more important is the exposure of the PAP south of the present coast during glacial stages. Especially when sea level sank below -100 m (Figure 1), a large area for colonization and growth of thicket, woodland on the floodplain, Renosterveld and Fynbos
was exposed (Cowling et al., 2020). This is suggested by pollen percentages of the Asteraceae pp (r=0.34, p<0.01) and Ericaceae (r=0.58, p<0.01) correlating significantly with sea level (Bintanja et al., 2005) (Figure 6, Table 2). Rapid sea-level rise during Terminations would have swamped the vegetation on the Agulhas bank and eroded the soils, which offers an alternative explanation for the subsequent pollen maxima of *Anthospermum*, Aizoaceae and *Kiggelaria-Spirostachys* type during Termination II (Figure 5).


**Table 2. R-values for linear correlations between concentration per ml of micro-charcoal particles with concentrations of selected pollen taxa; between pollen percentages and foraminiferal stable oxygen isotopes as well as different possible forcing mechanisms such as the southern hemisphere summer latitudinal gradient (LIG), the southern hemisphere winter LIG and sea level. Correlations were calculated on equidistantly interpolated values resampled every 3 ka between 5 and 305 ka. R-values corresponding to a p-**
**value < 0.05 are underlined, those corresponding to a p-value < 0.01 in bold. The two right columns indicate the periodicities in which power of the accumulation rates of selected pollen taxa (log transformed) exceeds the 90% $X^2$ level (REDFIT spectral analysis). \* Curves shown in Figures 5 -7.**

| | r linear correlation | | | | | log(AR) power > 90% $X^2$ | |
|---|---|---|---|---|---|---|---|
| **Taxon** | conc. vs micro-charcoal | % vs summer LIG | % vs winter LIG | % vs sea-level | % vs $\delta^{18}O_{benthics}$ | periodicity precession band | periodicity obliquity band |
| **Aizoaceae pp** | -0.01 | **_-0.29_** | 0.15 | _-0.21_ | **_-0.52_** | NV | NV |
| *Pentzia-Cotula* **type** | _0.27_ | 0.00 | 0.03 | 0.10 | -0.05 | 19-25 | -- |
| **Amaranthaceae** | 0.20 | -0.19 | **_0.33_** | -0.17 | **_-0.39_** | 21-24 | -- |
| *Stoebe-Elytropappus* **type** | 0.20 | 0.15 | _0.25_ | 0.23 | 0.21 | *21-25 | *40 |
| *Anthospermum* | _0.26_ | -0.03 | 0.13 | **_0.27_** | 0.10 | 22-25 | -- |
| **Asteraceae pp** | **_*0.42_** | _0.24_ | 0.19 | **_*0.34_** | _0.34_ | 20-25 | -- |
| **Ericaceae** | **_*0.40_** | 0.09 | -0.01 | **_*0.58_** | _0.57_ | *20-25 | -- |
| **Podocarpaceae** | _0.21_ | 0.19 | **_*-0.39_** | _-0.34_ | -0.03 | *19-25 | -- |
| **Restionaceae pp** | **_*0.44_** | -0.05 | **_0.26_** | 0.06 | -0.11 | 22 | *37-40 |
| **Poaceae** | **_0.30_** | **_*-0.37_** | 0.14 | -0.06 | **_-0.27_** | 19-24 | 37 |
| **Cyperaceae** | **_0.31_** | **_-0.34_** | 0.08 | _0.22_ | 0.15 | 19-24 | -- |





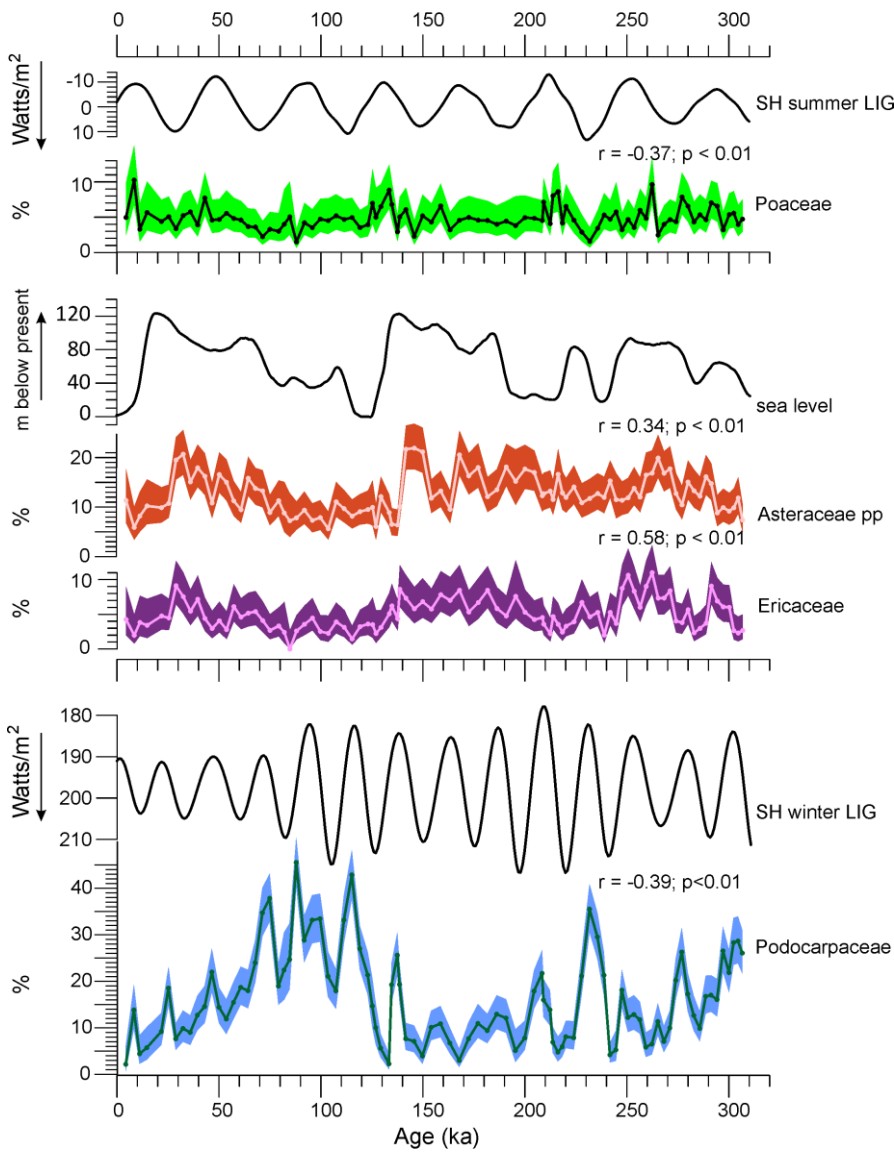

**Figure 6. Comparison of pollen percentages of Poaceae with the Southern Hemisphere summer latitudinal insolation gradient (SH summer LIG), of Asteraceae pp and Ericaceae with sea level (Bintanja et al., 2005), and of Podocarpaceae with the Southern Hemisphere winter latitudinal insolation gradient (SH winter LIG). Insolation gradients were calculated using the astronomical solution of La04 (Lascar et al., 2004). Note the reversed scales. Shadings indicate 95% confidence intervals.**

The modelled LGM vegetation on the PAP does not include forest (Cowling et al., 2020), perhaps because the climate and relatively flat geomorphology of the plain would favor rapid and expansive fires (Kraaij et al., 2020), precluding the growth of extensive forest on the plain. The low Podocarpaceae pollen percentages for the LGM corroborate this. However, we find a substantial increase in the representation of Podocarpaceae during MIS 7d and MIS 5 after the last interglacial maximum (MIS 5e) when sea-level was around 40 m lower than at present. Also, Esper et al. (2004) found substantial amounts of



bisaccate pollen (presumably from Podocarpaceae) corresponding to our Podocarpaceae maxima of MIS 5 in core GeoB3603-2 retrieved ca 15 km east of Site U1479. This indicates that the PAP might not always have been as open, as suggested by the LGM reconstruction. It is also possible that during phases of increased summer rain (see next section), Podocarpaceae extended in the Cape Fold Belt Mountains and the pollen were transported by rivers, in particular the Breede River. Wood charcoal assemblages from Elands Bay Cave prior to 20 ka gave evidence for several species characteristic for Afrotemperate forest,

including Podocarpus elongates, which were probably growing along rivers and streams (Cowling et al., 1999).

### 4.3  Open grasslands in the south-western Cape?

Pollen percentages for grasses are rather low (maximum of 10%, mean 5%), which is in line with glacial terrestrial pollen records of Cederberg Mountains and the south coast (Meadows and Sudgen, 1991; Scott and Woodborne, 2007a; 2007b; Quick et al., 2011, 2015; 2016; Valsecchi et al., 2013). Grass pollen is also relatively scarce at Vankervelsvlei and at Rietvlei grass

pollen is only abundant during the early Holocene (Quick et al., 2015; 2016). Modern values from the Namaqualand mudbelt along the west coast of South Africa run from ca 10% Poaceae pollen near Elands Bay adjacent to Fynbos vegetation to ca 40% in front of the Orange River, which drains the grass-rich landscapes of the South African interior (Zhao et al., 2016). Thus, the low Poaceae pollen percentages at Site U1479 are comparable to modern values derived from Fynbos and Renosterveld, and do not indicate extensive grasslands in the western part of the GCFR at any time during the past 300 ka.

In contrast to the pollen record, the Pleistocene fossil record of large-bodied herbivores from the GCFR has long been interpreted as indicating an expansion of grasslands, especially during glacial phases (Klein, 1983). South coast sites dating from MIS 6 onwards (e.g., Blombos Cave, Die Kelders Cave, Klasies River, Nelson Bay Cave, Pinnacle Point) frequently include extinct or extralimital grazers (e.g., Klein, 1972, 1976, 1983; Klein and Cruz-Uribe, 2000; Henshilwood et al. 2001; Rector and Reed, 2010), with very high abundances of grazers often observed during the LGM and late glacial (Klein, 1972,

1978, 1983; Faith, 2013). Stable isotope evidence ($\delta^{13}C$) from ungulate grazers at Boomplaas Cave indicate consumption of primarily $C_3$ grasses during the LGM (Sealy et al., 2016), as is also the case at Nelson Bay Cave (Sealy et al. 2020). Williams et al. (2020) measured stable isotopes on teeth of micromammals and macromammals from a hyena den found at Pinnacle Point and dated to MIS 6 (penultimate glaciation). The micromammal $\delta^{13}C$ indicate a $C_3$ vegetation in the immediate surrounding of the den, while the macromammal teeth indicate more mixed $C_3/C_4$ grasslands for the surrounding plain.

To reconcile the fossil pollen records to the fossil bone records from the south coast, one could hypothesize that the climate displayed a strong east-west gradient along the Palaeo-Agulhas plain from winter rainfall in the west to increased summer rainfall in the east. This pattern is indicated by the downscaling of the LGM results of coupled global climate model simulations to the region (Engelbrecht et al., 2019). Mapping of the glacially exposed PAP places the grasslands mainly in the central and eastern part of the plain with Fynbos and Renosterveld vegetation dominating the western part during the LGM (Cowling et

al., 2020; Marean et al., 2020). It is envisaged that the PAP would have harboured grazers migrating from the west in winter to the east in summer and back to make optimal use of grazing opportunities between winter rainfall and summer rainfall regions (Marean, 2010; Faith and Thompson, 2013; Copeland et al., 2016). However, a test of this proposed migration using



isotopic measurements of serially-drilled herbivore teeth from Pinnacle Point could not definitively corroborate a seasonal migration pattern (Hodgkins et al., 2020).

In this scenario, the pollen assemblages at Site U1479 would mainly register the vegetation of Renosterveld, Fynbos and riparian woodlands on the PAP. However, this concept is inconsistent with fossil evidence of micro- and macromammals indicating that the latest Pleistocene witnessed an expansion of open grassland relative to modern times in the south-western Cape as well (Avery, 1982; Klein and Cruz-Uribe, 1987; 2016). Glacial faunal assemblages document numerous grazers and open-country species (e.g., springbok, wildebeest, white rhino, quagga). For instance, Elands Bay Cave (Klein and Cruz-Uribe,

1987; 2016), Swartklip 1 (Klein, 1975), Hoedjiespunt 1 (Klein, 1983), and Die Kelders Cave (Klein and Cruz-Uribe, 2000) all indicate abundant grazing fauna feeding mainly on $C_3$ grasses (Sealy et al. 2016, 2020). Also, the LGM pollen spectra from Elands Bay Cave show strongly increased grass pollen percentages (Parkington et al., 2000). Perhaps a grassy under-story, such as those often associated with the Renosterveld, were sufficient to support these grazers. Alternatively, the vegetation of the western portion of the PAP consisted more of a mixture between grassland, Fynbos and Renosterveld. However, the modern

vegetation of the south-western Cape does not support grazers, such as mountain zebra (*Equus zebra*), very well (Faith, 2012). The grass species composition in pre-Holocene Renosterveld may have differed from the modern situation (and was possibly more palatable). Unfortunately, grass pollen cannot be determined to species level so it is not possible to assess the specific grasses available to these grazers. We could also speculate that grazing pressure might have subdued flowering and pollen production of the grasses, or that reduced atmospheric $CO_2$ enhanced grass quality (lower C:N ratio) and supported greater

grazer biomass (e.g., Owensby et al., 1996). On the other hand, the grasslands of the western portion of the PAP might be underrepresented in the pollen records, because other plants in the vegetation, such as Restionaceae, produced more pollen. This may result in a pollen record that underestimates the extent of grasses in these ecosystems. A final possible explanation for the discrepancy between the pollen and bone records is that Pleistocene grazers may have browsed more than their Holocene counterparts (e.g., Stynder, 2009).

**4.4   Vegetation associated with fire**

Elevated micro-charcoal concentrations seem associated with Fynbos vegetation (Figure 7, Table 2). Significant positive linear correlations (p<0.01) were found between micro-charcoal and pollen concentrations of Asteraceae pp (r=0.42), Restionaceae (r=0.44) and Ericaceae (r=0.40), but not with *Stoebe-Elytropappus* type. Micro-charcoal and *Pentzia-Cotula* type concentrations show a weaker correlation, however this is still significant (r=0.27, p<0.05). There is a slight correlation with

Podocarpaceae (r=0.21), but not with Amaranthaceae nor Aizoaceae pollen concentrations. The micro-charcoal record indicates that, as expected, fires were more frequent in asteraceous and ericaceous Fynbos vegetation but maybe less frequent in the Renosterveld (Mucina and Rutherford, 2006). Minimum micro-charcoal concentrations occurred during the interglacials MIS 7c, MIS 5e and MIS 1, and were similarly reduced during MIS 8 and MIS 4. Quick et al. (2016) found lower charcoal concentrations corresponding with a better representation of thicket species at Vankervelsvlei associated with decreased

seasonality.



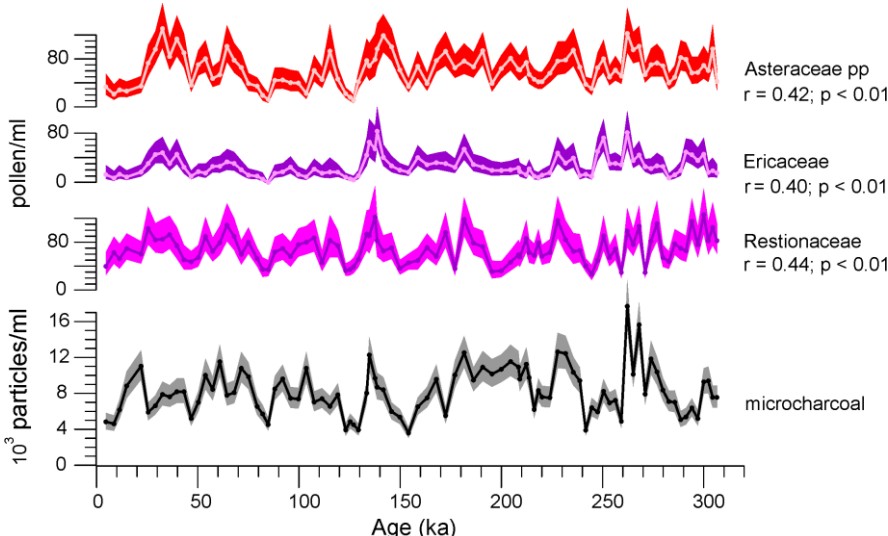

**Figure 7. Comparison of micro-charcoal particle concentration with pollen concentrations of Asteraceae pp, Ericaceae and Restionaceae. Shadings indicate 95% confidence intervals.**


### 4.5 Orbital forcing

The spectral analysis results indicate strong influence of precession and some influence of obliquity (Tables 2 and 3; Figure 8). We use here the SH winter LIG as an expression of the precessional variability. The precession component in the pollen accumulation rates is not an artefact of the age model based on tuning of the colour data as the age-model derived sedimentation

rates are rather constant. Moreover, spectral analysis of pollen percentages which are independent of changes in sedimentation, also reveals precessional variability (not shown, but see the correlation between Podocarpaceae pollen percentages and the SH winter LIG in Figure 6). Cross-spectral analysis indicates a negative correlation between the SH winter LIG and the accumulation rates of fern spores and pollen of *Pentzia-Cotula* type, Podocarpaceae, *Stoebe-Elytropappus* type, Ericaceae, Cyperaceae, Poaceae pp, Asteraceae pp, and Restionaceae (Table 3).

The precessional forcing might be associated with the SH winter LIG, which forces the latitudinal temperature gradient (Bosmans et al., 2015; Zhao et al., 2020). Partridge et al. (1997) previously proposed precessional forcing of rainfall in eastern South Africa. Only more recently, studies of independently dated marine sediments from the western Indian Ocean confirmed the importance of precessional forcing on the discharge of southeast African rivers, such as Limpopo and Tugela Rivers (Simon et al., 2015; Caley et al., 2018). It should be noted that SST of the Agulhas waters in the western Indian Ocean do not show a

precessional rhythm (Caley et al., 2011; 2018). Sensitivity modelling using maximum and minimum precession showed that austral summer precipitation in southeast Africa is higher when precession is at maximum resulting in maximum austral summer insolation (Bosmans et al., 2015; Simon et al., 2015) increasing the extension of the SRZ and probably also of the YRZ. This in turn corresponds to minima in the SH winter LIG (i.e., the difference in insolation between high- and mid-





southern latitudes during austral winter). The model experiment also indicates that maximum precession, i.e., minimal SH

winter LIG, induces weakening of the Southern Hemisphere westerlies (Simon et al., 2015).

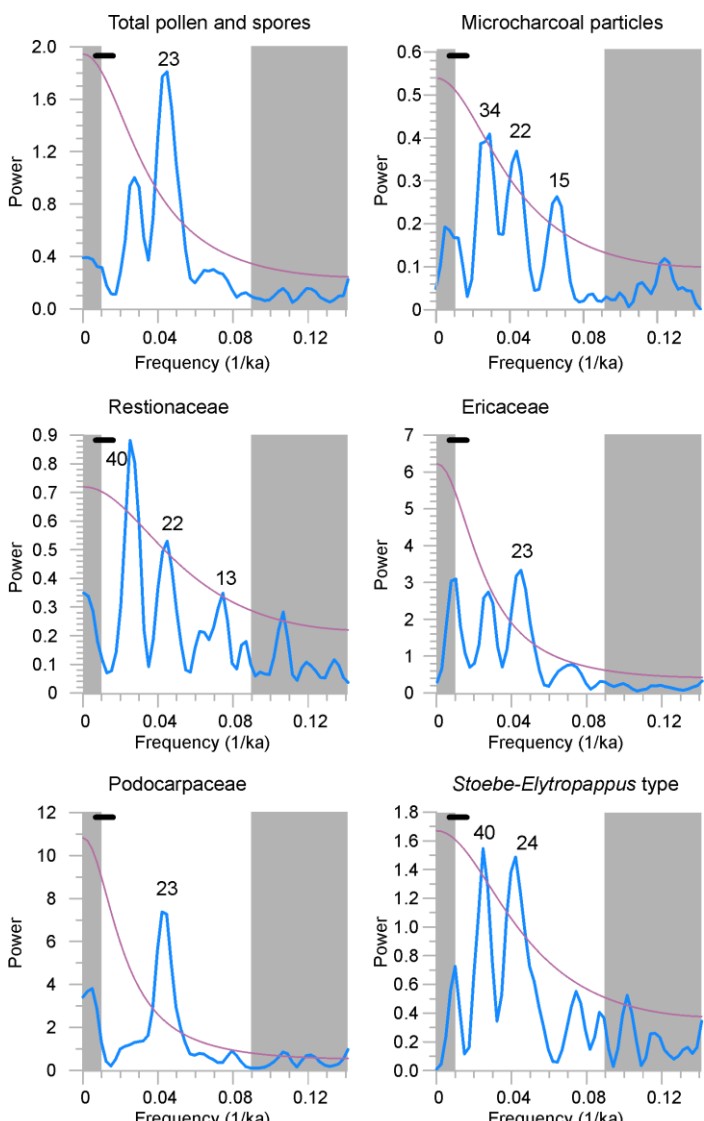

**Figure 8. Spectral analysis of the logarithmic transformed accumulation rates of total pollen and spores, micro-charcoal particles and pollen of Restionaceae, Ericaceae, Podocarpaceae and *Stoebe-Elytropappus* type using REDFIT (Schulz and Mudelsee, 2002;**

**PAST, Hammer et al., 2001). Lilac lines denote the 90% X² significance level. Bandwidth is indicated by the bars top left of each panel. Grey shading obscures the spurious results for those frequencies that are too high for the resolution or too low for the length of the record.**

Conversely, strong SH winter LIG during precession minima could induce strong westerlies resulting in more precipitation,

intensified seasonality and an extension of the WRZ at the cost of the YRZ along the south coast. Increased seasonality might





have been unfavourable for the growth of Podocarpaceae, explaining the significant negative correlation (r=-0.39, p<0.01) of Podocarpaceae values with the SH winter LIG (Figure 6, Table 2). Increased river discharge of rivers in the YRZ during maximum precession could explain the precessional variability in the accumulation rates of other taxa (Table 2). For variability in ocean conditions, Esper et al. (2004) found strong precessional variability in the dinoflagellate cyst record of core

GeoB3603-2 (located ca 15 km east of Site U1479) indicating that stratified oligotrophic waters prevailed when precession was weak (SH winter LIG strong).

**Table 3. Results of the Blackman-Tukey cross-spectrum analysis (3-306 ka, step 3 ka, 40 lags, bandwidth 0.01111 ka-1) executed with ARAND software (Howell et al. 2006). Phase is given for SH winter LIG multiplied by minus 1 against the log(AR) in cm² ka⁻¹**

**for fern spores, selected pollen taxa, total pollen and spores, micro-charcoal. 95%-level of non-zero coherency is 0.84 (denoted in bold); 80%-level of non-zero coherency is 0.70. These taxa exhibit a negative correlation with SH winter LIG lagging by up to 2.6 ka.**

| Taxon log(AR) | coherency | phase (°) | error (°) | lag (ka) | error (ka) |
|---|---|---|---|---|---|
| | | **(SH winter LIG)x(-1)** | | | |
| **fern spores** | **0.84** | **41** | **17** | **2.6** | **1.1** |
| *Pentzia-Cotula* **type** | 0.78 | 6 | 22 | 0.4 | 1.4 |
| **Podocarpaceae** | **0.99** | **23** | **3** | **1.5** | **0.2** |
| *Stoebe-Elytropappus* **type** | **0.95** | **42** | **10** | **2.6** | **0.6** |
| **Ericaceae** | **0.86** | **24** | **16** | **1.5** | **1** |
| **Cyperaceae** | **0.89** | **20** | **15** | **1.5** | **0.9** |
| **Poaceae pp** | 0.76 | 34 | 23 | 2.2 | 1.5 |
| **Asteraceae pp** | **0.87** | **11** | **15** | **0.7** | **1** |
| **Restionaceae** | **0.91** | **6** | **12** | **0.4** | **0.8** |
| **total pollen and spores** | **0.95** | **24** | **9** | **1.6** | **0.6** |
| **micro-charcoal** | **0.86** | **18** | **16** | **1.1** | **1.1** |

Accumulation rates of Podocarpaceae pollen show highest coherency with SH winter LIG and the maximum in accumulation

rates occurs $1.5 \pm 0.2$ ka after the minimum of SH winter LIG (phase lag of $23° \pm 3$ between precessional forcing and vegetation response, Table 3). Our results are in line with a transient simulation of monsoon climate over the past 280 ka focusing on precessional variability (Kutzbach et al., 2008). This simulation indicated that the South African monsoon (in the SRZ) responded with a phase lag of slightly less than one month (30°) to maximum December insolation, which is very close to the response of Podocarpaceae pollen accumulation rates.

Obliquity forcing could be explained by the latitudinal temperature gradient during summer (Davis and Brewer, 2009) that can be estimated by the SH summer LIG. Cyperaceae and Poaceae accumulation rates show power at obliquity periods while Poaceae and Cyperaceae pollen percentages correlate negatively with SH summer LIG (Figure 6, Table 2). Also, pollen accumulation rates of Restionaceae and *Stoebe-Elytropappus* type values show significant power at the obliquity band.





However, Ericaceae and other Fynbos related elements only show significant precession variability hinting at a heterogeneous

response of different vegetation types in the GCFR.

### 4.6    Comparison with the Piacenzian record

In a separate study, Zhao et al. (2020) described the late Pliocene (Piacenzian) vegetation development of south-western Africa based on pollen, micro-charcoal and benthic foraminifera oxygen isotope records of the same IODP Site U1479. We compare the two periods; the period from 3337 to 2875 ka including the mid-Piacenzian Warm Period (mPWP, 3264-3025 ka) with the

period of this study (307-0 ka). The pollen assemblages of both periods show continuity in the floristic composition although differences in the relative abundance are obvious (Figure 9).

The most striking differences between the pollen assemblages are changes in the abundances of Podocarpaceae and Restionaceae (Figure 9, Supplementary Figure 4). While during the Piacenzian, Podocarpaceae pollen percentages range between 1% and 15% with a median of 4.5%, maximum values and variability are much higher during the past 300 ka (2-46%,

median 14.4%). Conversely, the percentages for Restionaceae are much higher in the Piacenzian (18-47%, median 33.2%) than in the Pleistocene (7-27%, median 14.6%). Other changes in the assemblages are less dramatic but still important. Aizoaceae pp. (excluding *Ruschia* type) pollen has prominent values during the Pleistocene but hardly occurs during the Piacenzian. *Zaluzianskya* type pollen occurs continuously in the Pleistocene, reaching maximum values of 2% but its regular appearance started only after 3060 ka in the Pliocene record. On the other hand, *Protea* pollen is much more abundant during

the Piacenzian. Finally, maximum values and variability for *Anthospermum*, *Stoebe-Elytropappus* type, Asteraceae and Ericaceae increased in the Pleistocene.

The decline in the representation of Restionaceae and the increase of Podocarpaceae after the Pliocene indicate shrinking of the WRZ and the area with Fynbos. Increase in Aizoaceae and *Zaluzianskya* type suggests some expansion of the Succulent Karoo. Generally cooler and possibly drier conditions of the Pleistocene could have induced these shifts in vegetation.

Extension or westward shift of the YRZ could have allowed Podocarpaceae to spread into the south-western Cape, where it nowadays occupies pockets in the Cape Fold Belt Mountains (Mucina and Geldenhuys, 2006).

Zhao et al. (2020) interpreted the Piacenzian pollen record as the result of climate change in relation to SST and the SH winter LIG. The combination of a strong latitudinal insolation gradient inducing strong southern westerlies and warm SST would have increased precipitation in the WRZ during the Piacenzian. Conversely, a weak insolation gradient and weak westerlies in

combination with cool SST would have decreased the winter precipitation and reduced the extension of the WRZ along the south coast. These climatic conditions would have shifted the composition of the Fynbos vegetation to more Ericaceae-rich and increased the Afrotemperate forest (Podocarpaceae) in the YRZ. These mechanisms probably were still effective in the Pleistocene and suggest that the WRZ was reduced during MIS 4 and the cooler periods of MIS 5. Increased glacial-interglacial variability, decreased SST (especially in the upwelling regions) and low atmospheric $CO_2$ during glacial periods further

affected the vegetation (Dupont et al., 2019). Restionaceae became less abundant and Podocarpaceae forest extended during specific periods such as MIS 5d-a.





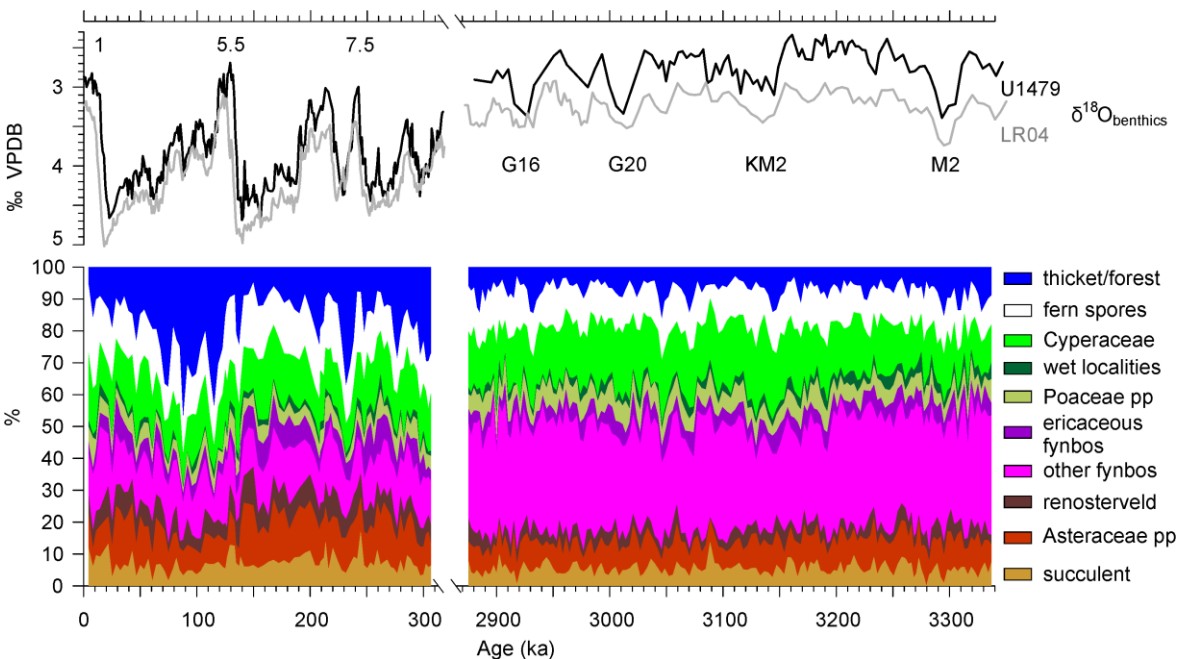

**Figure 9. Summary diagram of the Pleistocene (left) and the Pliocene (right) pollen record of Site U1479. Note the larger fluctuations in the Pleistocene and the strong increase of thicket/forest (mainly Podocarpaceae) pollen, increase in Asteraceae pp pollen, a slight increase in succulent pollen, and decline in other Fynbos (mainly Restionaceae) pollen compared to the Pliocene. Pollen groups are listed in Table 4.**

Not only did variability and relative abundance of pollen taxa change, but also the rhythm in the variability differs between the Piacenzian and Pleistocene. Spectral analysis of percentages, concentration and accumulation values of the most important pollen taxa showed precessional (24-18 ka) variability but no power in the obliquity band (40-42 ka) during the Piacenzian (Zhao et al., 2020), whereas the Pleistocene record shows significant power at the obliquity period in the pollen accumulation rates of Restionaceae, Cyperaceae and Poaceae (Table 2).

**Table 4. Groups of pollen taxa in Figure 9.**

| | |
|---|---|
| **Ericaceous Fynbos** | Ericaceae, Thymelaeaceae (incl. *Passerina*) |
| **Other Fynbos** | Bruniaceae, Proteaceae, Restionaceae, Rhamnaceae, |
| **renosterveld** | *Stoebe-Elytropappus* type, *Cliffortia*, *Anthospermum* |
| **Succulent and drought adapted** | Aizoaceae, Amaranthaceae, Crassulaceae, *Euphorbia*, *Zaluzianskya* type *Tribulus Pentzia-Cotula* type |
| | *Rhus* type, *Dracaena*, *Buxus*, Celastraceae Combretaceae Ebenaceae Euphorbiaceae pp, *Kiggelaria-Spirostachys* type, *Morella*, *Olea*, Podocarpaceae, *Myrsine*, *Dodonaea viscosa*, Sapotaceae, *Celtis*, *Ilex*, *Pycnanthus*, *Syzygium* type, Anemiaceae, Polypodiaceae, *Pteris-Cyathea* type |
| **Thicket/forest** | *Cussonia*, *Anthocleista*, *Drosera*, *Gunnera*, *Myriophyllum*, *Nuphar*, *Plantago*, *Phragmites* type, |
| **Wet localities** | *Oxygonum-Polygonum*, *Clematis* type, *Solanum*, *Typha*, Urticaceae |





The appearance of the obliquity rhythm in the Pleistocene pollen record might be an effect of the much larger climate change between glacial and interglacial periods, but could also be the result of vegetation on the PAP during periods of low sea level during the Pleistocene. During these low sea-level periods, especially when global sea-level dropped below 100 m, exposed regions became available for Asteraceae-rich vegetation such as Renosterveld, thicket and woodland on the floodplains. Also, drought-adapted pioneers might have found suitable habitats on the PAP during Pleistocene glaciations. The dramatic environmental changes that must have accompanied the flooding of the PAP during Terminations would have induced both erosion and changes in the vegetation reflected in the pollen record.

## 5  Conclusions

Deep-sea sediments from IODP Site U1479 retrieved from SW of Cape Town provide a 300-ka long record of pollen, spores and micro-charcoal from the Greater Cape Floristic Region indicating developments in Fynbos, succulent vegetation, thicket and forest over the course of three glacial-interglacial cycles. The micro-charcoal record suggests that fires occurred more in Fynbos than in other vegetation types of the GCFR. Our data indicate a substantial change in vegetation during the Terminations of glacial periods. During sea-level low-stands, the vegetation on the western part of the PAP would have consisted of asteraceous and ericaceous Fynbos including Renosterveld as well as thicket and woodland. Our results corroborate the vegetation modelling for the PAP during the last glacial maximum. However, we also infer extension of Podocarpaceae forest during parts of MIS 7 and MIS 5-4. The record exhibits relatively little grass pollen, which is difficult to reconcile with fossils of grazers found at Pleistocene paleontological and archaeological glacial sites across the GCFR.

Precession variability of the Podocarpaceae record, in particular, indicates increased summer rain and extension of the year-round rainfall zone in southernmost Africa at the cost of the winter rainfall zone during phases of maximum precession, when local summer insolation was high and the latitudinal insolation gradient in austral winter was weak. While the pollen accumulation rates of most taxa only exhibit a precessional rhythm, those of Poaceae, Restionaceae and *Stoebe-Elytropappus* type show both precession and obliquity related variability. This differentiation suggests a heterogeneous response of different vegetation types in the GCFR to global climatic changes.

Comparison with the Pliocene (Piacenzian) record of the same site indicates that the Pleistocene-Holocene winter rainfall zone decreased in size compared to the Pliocene. Reduced extension of the winter rainfall zone along the south coast could have been the result of weak westerlies in combination with cool SST, which decreased precipitation during winter. However, insolation forcing and precession variability remained drivers of rainfall distribution and seasonality. The large sea-level fluctuations of the late Pleistocene intermittently exposed the Palaeo-Agulhas Plain, but seemingly did not alter the rainfall patterns.



**Acknowledgements**

The study was financially supported by the Deutsche Forschungsgemeinschaft (DFG grant DU211/7) and the National Science Foundation (Award #1826666). LMD thanks Nikolaos Ioannidis for his help with the palynological preparations.

**Author contributions**

LMD designed the study, carried out the palynological analysis and wrote the draft, CC carried out the isotope analysis and constructed the age model. All authors contributed to the discussion and the final manuscript.

**Data availability**

Data are archived in PANGAEA.de ([https://doi.pangaea.de/10.1594/PANGAEA.930614](https://doi.pangaea.de/10.1594/PANGAEA.930614))

**Appendix**

Supplementary Figure 1. Correlation of upper cores of Holes C, B, G, H and I of IODP Site U1479 using the green channel of the colour scan (Hall et al., 2017 and Shipboard data). On top the stable oxygen isotopes of *Planulina wuellerstorfi*. VPDB, Vienna standard PeeDee Belemnite.

Supplementary Figure 2a. Percentages of pollen and spores for selected taxa from Fynbos and drought-adapted vegetation. On top the stable oxygen isotopes of benthic foraminifera of Site U1479 (black) and global stack LR04 (grey, Lisiecki and Raymo, 2005). Zones after stratigraphically constrained cluster analysis (PAST, Hammer et al., 2001). VPDB, Vienna standard PeeDee Belemnite.

Supplementary Figure 2b. Percentages of pollen and spores for selected taxa from forest, thicket and wet localities. On top the stable oxygen isotopes of benthic foraminifera of Site U1479 (black) and global stack LR04 (grey, Lisiecki and Raymo, 2005). Zones after stratigraphically constrained cluster analysis (PAST, Hammer et al., 2001). VPDB, Vienna standard PeeDee Belemnite.

Supplementary Figure 3. Accumulation rates of selected pollen taxa and fern spores. On top the sedimentation rates. Zones after stratigraphically constrained cluster analysis (PAST, Hammer et al., 2001).

Supplementary Figure 4. Comparing Pliocene and Pleistocene pollen percentages of selected taxa. Shadings indicate 95% confidence intervals. On top the stable oxygen isotopes of Site U1479 (black, Zhao et al., 2020; this study) and the global stable oxygen stack LR04 (grey, Lisiecki and Raymo, 2005). Pliocene part after Zhao et al. (2020), Pleistocene part after this study. VPDB, Vienna standard PeeDee Belemnite. VPDB, Vienna standard PeeDee Belemnite.



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
