# Peer review of "Continuous vegetation record of the Greater Cape Floristic Region (South Africa) covering the past 300 thousand years (IODP U1479)"

_Climate of the Past, 2021_

## Author Comment (AC3)

Supplementary Table 1. *Correlations are carried out using the Prais-Winsten regression method (Hammer et al. 2001). Values in bold are significant after a Bonferroni correction (p<0.0009). The Breusch-Pagan test for heteroskedasticity, i.e. non-stationary variance of residuals, indicated that in most cases homoskedasticity could not be rejected. Exceptions in which homoskedasticity was rejected at the 5% level are* Poaceae *percentages vs summer LIG,* Podocarpaceae *and Stoebe-Elytropappus type percentages vs winter LIG, Anthospermum and Stoebe-Elytropappus type percentages vs sea-level. The residuals of all correlations failed the Durbin-Watson test for no positive auto-correlation. (see also reference manual for PAST vs 4:*
https://www.nhm.uio.no/english/research/infrastructure/past/downloads/past4manual.pdf)

| linear regression correlations | pollen concentrations against micro-charcoal concentration | | | pollen percentages against SH summer latitudinal insolation gradient | | | pollen percentages against SH winter latitudinal insolation gradient | | | pollen percentages against modelled global sea-level after Bintanja et al., 2005 | | | pollen percentages against global stable oxygen stack (LR04) | | |
|---|---|---|---|---|---|---|---|---|---|---|---|---|---|---|---|
| Taxon | r | p | residual p | r | p | residual p | r | p | residual p | r | p | residual p | r | p | residual p |
| Aizoaceae pp | -0.012 | 0.9060 | 0.8093 | -0,29 | 0.0036 | 0.0047 | 0.154 | 0.1235 | 0.1773 | -0.206 | 0.0352 | 0.3402 | -0.524 | **0.0001** | 0.0001 |
| *Pentzia-Cotula* type | 0.268 | 0.0104 | 0.9125 | 0,00 | 0.9844 | 0.8223 | 0.026 | 0.7932 | 0.3254 | 0.099 | 0.3230 | 0.6095 | -0.048 | 0.6416 | 0.5009 |
| Amaranthaceae | 0.198 | 0.0599 | 0.4826 | -0,19 | 0.0533 | 0.0650 | 0.331 | **0.0005** | 0.4625 | -0.166 | 0.0984 | 0.1929 | -0.392 | **0.0001** | 0.2753 |
| *Stoebe-Elytropappus* type | 0.201 | 0.0553 | 0.3268 | 0,15 | 0.1363 | 0.0748 | 0.253 | 0.0133 | 0.0457 | 0.232 | 0.0218 | 0.0283 | 0.210 | 0.0375 | 0.2902 |
| *Anthospermum* | 0.263 | 0.0141 | 0.1447 | -0,03 | 0.7493 | 0.4645 | 0.129 | 0.2022 | 0.5331 | 0.270 | 0.0072 | 0.0177 | 0.101 | 0.3108 | 0.6479 |
| Asteraceae pp | 0.424 | **0.0001** | 0.2699 | 0,24 | 0.0137 | 0.3312 | 0.190 | 0.0545 | 0.0756 | 0.336 | **0.0004** | 0.6874 | 0.341 | **0.0005** | 0.2800 |
| Ericaceae | 0.397 | **0.0001** | 0.2517 | 0,09 | 0.3586 | 0.1373 | -0.014 | 0.8864 | 0.9339 | 0.578 | **0.0001** | 0.8502 | 0.575 | **0.0001** | 0.2109 |
| Podocarpaceae | 0.208 | 0.0488 | 0.9597 | 0,19 | 0.0620 | 0.4400 | -0.387 | **0.0001** | 0.0242 | -0.341 | **0.0004** | 0.0699 | -0.028 | 0.7838 | 0.5246 |
| Restionaceae pp | 0.438 | **0.0001** | 0.7245 | -0,05 | 0.6075 | 0.1886 | 0.258 | 0.0090 | 0.1218 | 0.063 | 0.5273 | 0.6222 | -0.112 | 0.2602 | 0.0078 |
| Poaceae | 0.303 | 0.0036 | 0.0644 | -0,37 | **0.0003** | 0.0268 | 0.141 | 0.1623 | 0.5011 | -0.061 | 0.5472 | 0.1075 | -0.274 | 0.0054 | 0.0623 |
| Cyperaceae | 0.309 | 0.0038 | 0.2096 | -0,34 | **0.0001** | 0.3653 | 0.082 | 0.4096 | 0.0026 | 0.221 | 0.0253 | 0.8665 | 0.147 | 0.1385 | 0.3853 |

---

## Author Comment (AC4)

[Figure]

Supplementary Figure 5. Results stratigraphically constrained cluster analysis using unweighted pair-group average and correlation similarity index from the PAST package (Hammer et al., 2001). Zone I: 4-22ka; Zone II: 25-61ka; Zone IIIa: 64-123ka; Zone IIIb: 125-133ka; Zone IVa: 135-139ka; Zone IVb: 142-224ka: Zone V: 228-239ka; Zone VI: 242-271ka; Zone VII: 274-307ka.

---

## Author Response (AR1)

**Point-by-point reply to Reviewer #1**

**RC1:** Overall, I really enjoyed reading this paper and I am supportive of this high-quality manuscript, which fully deserved to be published in climate of the past once the following comments will be addressed.

**Response:** Thank you for your support. We answer you questions and address you comments point by point.

**RC1:** L100-101: Chase and Quick (2018) deal with time periods much longer than this. A reference presenting this modern dynamic would be more appropriate. Or the sentence could be rephrased to highlight that this behaviour has been shown to exist on longer timescales and could have a role on shorter timescale.

**Response:** We will rephrase the sentence and add the reference of Jury et al. 1993 (Jury, M.R., Valentine, H.R., Lutjeharms, J.R.E. Influence of the Agulhas Current on Summer Rainfall along the Southeast Coast of South Africa. Journal of Applied Meteorology, 32: 1282-1287).

*In addition, the Agulhas Current influences the climate of the coastal area (Jury et al., 1993), which on longer timescales could have propagated climate signals from the tropics to the southern Cape coast (Chase and Quick, 2018).*

**RC1:** L157: The equation represented on Supp Fig. 1 is wrong. It should be H1 * 6.76 / 6 (and not dividing by H1)

**Response:** Of course; thank you for spotting this error. We'll correct the figure. The depths, however, have been correctly calculated.

**RC1:** I think the chronological uncertainties could be potentially quite large. If I read figure 4 correctly, only 4 dates were used (only 4 changes of sedimentation rate) for a sequence of 300,000 years? I cannot understand is why each sedimentation rate interval seems to be about 75 kyr long if the tuning was based on precession. This doesn't fit with my understanding of a chronology based on precessional cycles and deserves some clarification.

And I would like the authors to also describe why/how 'sediment colour is demonstrably coherent with climatic precession'. This assumption seems key to the chronology, which looks very good in the end! So, this is enough evidence that the assumption was reasonable, but more details would be much appreciated.

**Response:** The reconstruction of the timescale of IODP U1479 using the orbital tuning approach covers the entire Plio-Pleistocene. It will be subject of a dedicated paper, that is still in preparation. The strong precessional cycles in colour were found throughout the sedimentary sequence, which extends to the late Miocene. Sediment colour depends largely on the carbon content, which variability could be the result of productivity in the surface ocean, dissolution in the deep ocean or dilution by terrestrial material. A full discussion about which effect is more important lies beyond the scope of the present paper. As the reviewer points out, the orbitally tuned colour chronology is close enough to the oxygen isotope chronology that our conclusions drawn from the pollen signal do not depend on choice of chronology. Because the correlation with the alternative splice used for the pollen analysis would introduce additional errors, we refrained from fine-tuning and only four control points were used in the range between 300 and 0 ka. We will insert more explanation in the section concerning the chronology as follows:

Lines 166-169 "Discrete shipboard measurements and XRF scanning indicate that the sediment colour essentially monitors variable carbonate content, but the orbital tuning approach requires no assumption about the actual mechanism through which orbital variability paces the carbonate variability." will be expanded into:

*Throughout the ca 6 Ma long record of Site U1479, sediment colour displays strong cyclical variability at frequencies associated with orbital (climatic) precession, and the discrimination of these cycles with depth suggests a modulation of amplitude similar to that of precession. Discrete shipboard measurements and XRF scanning indicate that the sediment colour essentially monitors variable carbonate content, but the orbital tuning approach requires no assumption about the actual mechanism through which orbital variability paces the carbonate variability. The most detailed orbital tuning typically requires either the manual anchoring of the ordinal points of every cycle or else the use of deterministic or probabilistic mapping techniques (e.g. Lin et al., 2014). However, under the circumstances, the transfer of such higher order assumptions from the shipboard splice to the alternative splice used for pollen analysis would introduce additional errors and, therefore, be of debatable merit. Thus, for this work, we simply adopt the minimum*

*number of chronological anchor points necessary to achieve significant correlation between sediment colour and precession.*

*Lin, L., D. Khider, L. E. Lisiecki, and C. E. Lawrence (2014), Probabilistic Sequence Alignment of Stratigraphic Records, Paleoceanography 29, 976-989, doi:10.1002/2014PA002713.*

**RC1:** The methods section would gain clarity if some subsections were added.

**Response:** We'll divide the methods into three subsections: *Site location, composite depth, chronology; Sample preparation; Statistical methods.* The latter will be updated (see below).

**RC1:** L219: The pollen concentration curve is only represented on Supp. Fig 3 and not on Figure 5. The same applies to charcoal particle concentration.

**Response:** We'll omit the reference to Figure 5 concerning the pollen concentration. Figure S3 actually gives accumulation rates (i.e. concentration multiplied by sedimentation rate). The charcoal particle concentrations are depicted in the upper part of Figure 5, just below the bar with MIS.

**RC1:** The sedimentation rate is fairly constant, while the rate of pollen and spore deposition varies hugely during the same period (factor 2 or 3 vs. a factor of 10+). How could more pollen grains be brought to the site without additional sediments? More pollen is produced during specific periods? Change of source? A quick word about this would tie everything nicely.

**Response:** The sediments at Site U1479 are mostly pelagic and the terrigenous component is low but variable (quartz: 6 ± 5%; clay: 8 ± 2%). Thus, terrestrial input is small compared to the marine material and fluctuations in the terrigenous component do not dominate the sedimentation, but variation in terrigenous input can lead to substantial variation in pollen deposition. In the discussion section of the manuscript, we extensively discuss the variability of the pollen record. We don't think a quick word in the results would be appropriate.

We'll start the paragraph at line 132 adding:

*The sediments at Site U1479 are mostly pelagic and the terrigenous component is low but variable (quartz: 6 ± 5%; clay: 8 ± 2%).*

**RC1:** L227-228: I am confused by how (and possibly why) this log transformation was applied. The authors argue here that it is to limit the effect that all percentages must sum to 1 (or at least this is my understanding). But the results are presented on Supp. Fig. 3 as counts / m2 / kyr, i.e. the different AR are independent. In this context, my question is: why the log?

**Response:** To avoid the interdependence introduced by the percentage calculation, we used the accumulation rates (with or without log transformation). However, the numbers in the accumulations rates are quite high. By applying a log transformation, the power results are expressed in much lower figures. We did run the analyses also on accumulation rates without log transformation, but the results were essentially the same.

We will change lines 227-228 to:

*We performed spectral analysis on the accumulation rates (AR) to avoid interdependence between the data as is the case with percentages. A log transformation [log(AR)] allows for the comparison of variables that spread across several order of magnitude on a comprehensible scale.*

**RC1:** I am also having a bit of a hard time following what data are used to create Table 2 and which ones are used in Figs. 5 to 7. Were the log(AR) or the AR used for the correlations? Or is it the percentages, since the authors suggest the data are plotted on Figs. 5 to 7. I think a clarification of all these elements are necessary to ensure that the reader can be certain o follow which data are used when and why.

**Response:** The headings of Table 2 are somewhat scrambled up and we agree that the caption is not too clear. We used AR data only in the spectral analysis. We will also publish the p-values in a supplementary table and follow up the suggestion to apply a Bonferroni correction (see next comment). The caption of Table 2 should be:

*Table 2. R-values for linear correlations between concentration per ml of micro-charcoal particles with percentages of selected pollen taxa (1. column), between pollen percentages and foraminiferal stable oxygen isotopes (5. column) as well as between pollen percentages and different possible forcing mechanisms such as the Southern Hemisphere summer latitudinal insolation gradient (LIG) (2. column), the Southern Hemisphere winter LIG (3. column) and sea level (4. column, Bintanja et al., 2005). Correlations were calculated using pairwise regression analysis on equidistantly interpolated values resampled every 3 ka between 5 and 305 ka. We applied a Bonferroni correction and r-values corresponding to a critical p-value < 0.0009 (0.05/55) are denoted in bold. Columns 6 and 7 indicate the periodicities in which power of the accumulation rates of selected pollen taxa exceeds the 90% $X^2$ level (REDFIT spectral analysis). * Curves shown in Figures 5 -7.*

**RC1:** Table 2: Then I have a problem with Table 2. Serial correlation approaches (i.e. when one record is repeatedly compared with other records) require the p_values to be adapted to the risk of false positive (see for instance section 3 of www.doi.org/10.1016/j.epsl.2016.11.048 or any other references dealing with the topic). For instance, it is certain that the p_values that are presented as significant at the 0.05 but not 0.01 thresholds (underlined not bold) will not resist a basic Bonferroni correction. Depending on the p_values of the bolded and underlined values, some of them risk to also lose their significance once corrected. This must be accounted for, and the table corrected accordingly. The advantage of doing this is that it will more clearly differentiate the strongest relationships from the background noise.

**Response:** We'll follow your suggestion to apply a Bonferroni correction. The correlations in Table 2 are calculated using regression analysis of pairs and will be better described in the method section of the new version. A supplementary table with p-values will be added.

[revised manuscript text omitted]

Accumulation rates of Podocarpaceae pollen show highest coherency with SH winter LIG and the maximum in accumulation rates occurs 1.5 ± 0.2 ka after the minimum of SH winter LIG (phase lag of 23° ± 3 between precessional forcing and vegetation response, Table 3). Our results are in line with a transient simulation of monsoon climate over the past 280 ka focusing on precession variability (Kutzbach et al., 2008). This simulation indicated that the South African monsoon (in the SRZ) responded with a phase lag of slightly less than one month (30°) to maximum December insolation, which is very close to the response of Podocarpaceae pollen accumulation rates.

Apart from precession variability our results also indicate some influence of obliquity. Obliquity forcing could be explained by the latitudinal temperature gradient during summer (Davis and Brewer, 2009) that can be estimated by the SH summer LIG. Cyperaceae and Poaceae accumulation rates show power at obliquity periods while Poaceae and Cyperaceae pollen percentages correlate negatively with SH summer LIG (Figure 6, Table 2). Also, pollen accumulation rates of Restionaceae and Stoebe-Elytropappus type values show significant power at the obliquity band. However, Ericaceae and other Fynbos related elements only show significant precession variability hinting at a heterogeneous response of different vegetation types in the GCFR.

**RC1:** Many taxa seem to also have an eccentricity component to their variability (Podocarpus, Asteraceae, Ericaceae and possibly more). I think these are important features that are a bit lost in comparison to the role of precession.

**Response:** The eccentricity component in the mentioned curves might be an effect of the precession related variability, as the amplitude of precession is modulated by eccentricity.

**Point-by-point reply to Reviewer #2**

**Urrego:** Dupont et al present a novel record of vegetation change and fire activity from the Greater Cape Floristic Region that spans over 300 ka. The paper provides insight into the development of vegetation in the southern tip of the African continent, from a site under the influence of major oceanic and atmospheric systems relevant not only for the understanding of environmental change in Southern Africa, but also for understanding of the global climate system. The authors develop a chronology that is not only dependent on the global isotope stack, giving the record some independence. The paper also includes a strong statistical treatment of pollen data and other independent variables that should be praised. I recommend the paper for publication in Climate of the Past as my suggestions are largely of format.

**Response:** Thank you for your positive assessment. We answer you questions and address you comments point by point.

**Urrego:** The authors mention that 'existing paleo-environmental records (from the GCFR) do not encompass a full glacial interglacial cycle'. This is inaccurate and should be modified in the abstract and introduction. Instead, the pollen and charcoal record of site MD96-2098 should be incorporated as one existing record of vegetation and fire change in the GCFR also covering two full glacial cycles (Daniau et al 2013 PNAS, Urrego et al 2015 Climate of the Past). The charcoal record from MD96-2098 (Daniau et al 2013) seems particularly relevant as this is the first paper to test and discuss the major influence of precession on fire activity in the GCFR. The author's results from IODP site U1479 should therefore be put in the context of these earlier findings. Additionally, the authors will find that the pollen calibration presented in Urrego et al 2015 Climate of the Past is highly relevant to this research and could incorporated in the interpretation of pollen signals from the GCFR and their ecological grouping.

**Response:** In this manuscript we focus on the vegetation of the western and southern Cape of South Africa. We discus the record of Site U1479, which covers a region situated much further south than the one covered by the record of Site MD96-2098. The latter is located at 25°36'S, i.e. almost 10 degrees of latitude north of Site U1479. In addition to the pollen blown from Namibia, sediments at Site MD96-2098 probably record the vegetation in the region of western South Africa and southern Namibia drained by the Orange River. Although pollen from Fynbos elements have been found in the Marion Dufresne core, we don't think it can be considered a record of the GCFR vegetation. A discussion about the variability of the Cape Flora vegetation in relation to the variability in the desert and savanna regions of southern Africa is beyond the scope of the current paper.

**Urrego:** In section 1.1 Modern climate and vegetation, the authors present a large amount of information that should be supported by primary literature. Between lines 104 and 109 the authors use barely any citation, which bears the question about the source of this information. Precipitation ranges and species composition of vegetation types are described but they lack scientific sources.

**Response:** In the description of the modern vegetation we refer to several chapters of Mucina and Rutherford (2006), the standard work about South African vegetation. To be more explicit, we'll add the reference of Rebelo et al. (2006) on line 108.

**Urrego:** The methods section and statistical analysis description should include a justification for choosing the taxa used for the correlation analysis and presented in Table 2. There are taxa that show significant correlations in Table 2 but are not plotted in the figures. It is not clear how the decision of what pollen taxa would be used in these analysis has been made. The same applies to the taxa shown and presumably chosen for the Spectral analyses and Black-Tukey cross-spectrum analysis. Again, there should be a justification in the methods for choosing these taxa.

**Response:** We selected the taxa based on the possible ecological meaning paired to sufficient occurrence of specific pollen grains to run meaningful statistics on the results.

**Urrego:** The section 4.1 Source area and pollen transport should be moved to the methods and environmental setting part of the paper. Here it is presented as part of the discussion, where the narrative is in danger of falling in a circular argument. First it is established that the pollen assemblage represents the nearby continental vegetation and later the pollen assemblage is used to reconstruct the composition of the

continental vegetation. When this section is incorporated into the methods, it should be phrased in a way that establishes the pollen sources more independently. Another point that should be incorporated in this section of pollen sources is the potential effect of the Agulhas leakage on the pollen record and the recorded vegetation. Is it possible that at some points during the Late Pleistocene, the pollen record may have originated from vegetation growing further East and carried over to the core site by oceanic currents? This is probably not the case, but it should be explicitly discussed to pre-empt reservations from the reader.

**Response:** Here, we do not agree with the reviewer. The section about pollen transport definitively belongs in the discussion and not in the methods section. We argue that the pollen flora represents the vegetation of the nearby continent, Africa, in contrast to hypothetical wind transport from South America. It is very well possible that some pollen grains derived from East Africa. However, we presume that the influence of the nearby vegetation of the Cape dominates the pollen record found in sediments of U1479.

We'll change the first sentence of section 4.1 into:
*The floral composition of the palynological assemblage of Site U1479 indicates that it records the biomes of South Africa, in particular that of the GCFR, although limited pollen transport from the east coast cannot be excluded.*

It turned out to be difficult to estimate the quantitative role of the Agulhas Current in pollen transport to Site U1479, although we assume that its influence is considerable (lines 260-261). Emergence of the PAP might have reduced the influence of the Agulhas Current transport to Site U1479, as in that case pollen grains transported by the Breede River are likely more abundant. Although there is a minimum in the pollen concentration around 125 ka (max sea-level), we do not observe strongly increased pollen accumulation rates during periods of low sea-level.

**Urrego:** The pollen and fire record seem to be given less weight than results from previous work on vegetation modelling the Palaeo-Agulhas plain (PAP) and modelled sea level change. While the modelling results are very valuable, the direct nature of empirical data such as this pollen record should be recognised. For instance, the pollen concentration changes are subtle during the glacial periods, potentially suggesting that the modelled sea level change may be less drastic and comparable in magnitude between glacial cycles. A 100-m sea level decrease would have probably been recorded as a prominent increase in pollen concentration because vegetation would have been closer to the site. Does the pollen record suggest the global modelled estimates need to be tweaked for the south African region? Likewise, the pollen results suggest that grasslands were not as extensive in PAP as previously thought, but it seems like the authors are attempting to find an interpretation that still fits the modelled vegetation of PAP. A balance between the value given to information provided by this new record and previous modelling efforts should be attempted here.

**Response:** We do not exclude the possibility of glacial/interglacial climate variation influencing the vegetation of the Cape; see lines 294-296, 496. However, the exposure of such a large shelf area as the PAP in the vicinity of the site probably had the greater impact. We unexpectedly do not find a prominent increase in pollen concentration during periods of low sea-level, but we do record a different pollen assemblage during those times. We compare our results with the vegetation modeling of the PAP, but we also mention differences between our results and those of the modelling effort; see line 320 ff.

**Urrego:** The information presented from isotopes of mammal teeth suggests changes in the abundance of C3 and C4 plants but this is not discussed in light of this new pollen record. How do these compare? Is the pollen record adding some insight into the composition of the vegetation that has only been inferred from a spotty set of mammal fossil records?

**Response:** We do not understand this comment. A full section (4.3) is dedicated to the discrepancy between the mammal record (including isotope information) and the marine pollen record. Our discussion of isotope data is primarily concerned with C3 versus C4 grasses, which cannot be distinguished on the basis of their pollen (and is noted in the text)

**Urrego:** Figures 1 and 2, and all their panels, should be consistent in their geographical extend. They should all show the same latitude and longitude ranges for consistency and to allow more effective reading.

**Response:** The geographical extent of the panels should be obvious as all maps have coordinates. We

choose the size of the maps according to the structure we want to depict. Some features, such as the direction of ocean currents, extend over larger areas than others, such as the bathymetry of the Agulhas Bank.

**Urrego:** Figure 3. Indicate in the legend what the blue circles are.

**Response:** We'll add '*samples are denoted with blue circles*' in the caption of Figure 3.

**Urrego:** Figure 4. The differences between the oxygen isotopic record and the orbitally tuned chronology are said to be within error (lines 174-175). These errors should be shown in the Figure to support the statement.

**Response:** We refer to the estimated error for the LR04 timescale of ±4 ka for the past million years. We'll adapt the sentence on lines 174-175 as follows:
*the differences are generally within the error of ±4 ka for the past million years stated for the LR04 stack (PAGES, 2016), though there is isolated divergence in discrete intervals such as the penultimate deglaciation (Figure 4).*

**Urrego:** The results of the cluster analysis used for the zonation should be presented in the supplementary figure. At the moment they are quickly described in the text but not presented in a figure. It is hard to evaluate the validity of this zonation exercise without the complete results from the cluster analysis.

**Response:** Due to restricted space in the supplementary figures, we'll provide a separate supplementary figure with the cluster analysis results. We spotted a mistake in Table 1; Zone I includes 5 samples from 4-22ka and Zone II includes 11 samples from 25-61ka. We'll correct Table 1 in the final version.

**Urrego:** Figure 6. The sea level curve should be labelled "modelled (global?) sea level".

**Response:** We'll amend the sentence on line 301: *correlating significantly with modelled global sea level curve (Bintanja et al., 2005)*

**Urrego:** Table 2. Include citation for sea-level reconstruction in the legend. This is included in Figure 6 but not here.

**Response:** We'll adapt the caption of Table 2 (see also the response to RC1)

**Urrego:** Table 4. This table should include references on which the ecological grouping is based. It is also hard to read what taxa correspond to each ecological group. For example, is Cyathea type the only taxa included in the Thicket/forest? Or is this Pteris-Cyathea type and is grouped in the succulent and drought adapted? Horizontal lines separating each group could help avoiding confusion with this.

**Response:** We'll follow this suggestion and adapt the lay-out of Table 4. We'll add to the caption:
*Grouping follows Quick et al., 2015, 2016.*

---

## Author Response (AR2)

Point-by-point response by Lydie Dupont

- Explanation about the calculation of the summer and winter LIG is provided but it would merit more information about the parameters you used using Past software. Did you use the "daily insolation" or "insolation in a given month" for the curves presented in Fig. 6? I was able to reproduce the winter LIG using the month 21December-20January on Past software but not the summer LIG curve. Could you please provide detailed technical information about the parameters you used in Past software in the method section.

I did not use PAST to calculate the summer and winter LIG values but downloaded the daily insolation values for the different latitudes on June 21 and December 21 from the Lascar website and calculated the difference (with Excel) as stated on lines 207-209.

√ L261: typo "in particular"

- Figures 5 and 7: Microcharcoal and Restionaceae curves seem to be shifted between the two figures. For example, at the transition between MIS6/5 peak of microcharcoal is contemporaneous with a decline of Restionaceae on Fig. 5; whereas on Fig. 7 this peak of microcharcoal is contemporaneous with a peak of Restionaceae. Does this difference come from the use of different age models for drawing the two figures? Or does this difference come from the fact that you use pollen concentration on Fig. 7 whereas it is pollen percentages on Fig. 5?
If the difference comes from differences between pollen concentration and pollen percentages, could you please justify why pollen concentration is used to discuss microcharcoal concentration and fire, and why pollen percentages were not used? This is important as two different scenarios can be obtained between fire and vegetation (positive or negative relationships between fire and fynbos). Please adapt the text if necessary.

Figure 7 shows pollen concentration values to compare them with the micro-charcoal concentration. To compare pollen percentages with micro-charcoal concentration would be like comparing apples with oranges as the concentration values are influenced by the sedimentation rates and the percentages are not. Pollen concentrations of some taxa correlate well with the micro-charcoal concentration while others do not indicating that the correlation is not dominated by the sedimentation rates. In the revision of Table 2 an error crept in. The first column compares micro-charcoal with pollen concentrations, not —as erroneously stated— micro-charcoal concentration with pollen percentages. I apologize for that. The start of the caption will be changed as follows; **R-values for linear correlations between concentration per ml of micro-charcoal particles with concentration values of selected pollen taxa (1. column), ...**

- Microcharcoal concentrations (and the different taxa accumulation rate) are negatively correlated to the winter LIG (Table 3). It seems therefore that peaks in fire and Podocarpaceae are in phase with peaks in summer insolation at the latitude of the core, with precession maxima and with the summer rain in the Greater Cape Floristic Region (GCFR). As stressed by one of the reviewers in the first round of review, it is a pity that your interpretation of the pollen and microcharcoal records in core U1479 (GCFR) were not compared with the microcharcal and pollen record from core MD96-2098 (Daniau et al. 2013; Urrego et al. 2015). Those studies suggested that southern Africa was characterized by increasing summer rain at precession maxima. This is also what you observe in the GCFR. You compare your interpretation with records from the southeastern region (Partridge et al. 1997; Simon et al 2015 and Caley et al 2018). Including a brief comparison/discussion with results from core MD96-2098 (western part, Orange and probably part of Namibia basins) in the "Orbital forcing" section would complete the picture for the full region of southern Africa.

You are right, I'll include the results of MD96-2098 in the precession discussion, change lines 434-436 and 466, and update the reference list.

More recently, studies of independently dated marine sediments confirmed the importance of precessional forcing, whereby increased precipitation occurred during maximum precession; a micro-charcoal and pollen record from the eastern South Atlantic indicated precession forcing on the seasonality and amount of rainfall in the summer rainfall region of southern Africa (Daniau et al., 2013; Urrego et al., 2015), while records from the western Indian Ocean revealed precession-driven discharge of southeast African rivers, such as Limpopo and Tugela Rivers (Simon et al., 2015; Caley et al., 2018).

To line 466, I'll add: "In line with the mentioned inferences from marine sediments (Daniau et al., 2013; Urrego te al., 2015; Simon et al., 2015; Caley et al., 2018), our spectral analysis results also indicate..."

Daniau, A-L., Sánchez Goni, M.F., Martinez, P., Urrego, D.H., Bout-Roumazeilles, V., Desprat, S. & Marlon, J.R.: Orbital-scale climate forcing of grassland burning in southern Africa., PNAS, 110, 5069-5073. doi:10.1073/pnas.1214292110, 2013.

Urrego, D.H., Sánchez Goñi, M.F., Lechevrel, S. & Hanquiez, V.: Increased aridity in southwestern Africa during the warmest periods of the last interglacial., Climate of the Past, 11: 1417-1437, doi:10.5194/cp-11-1417-2015, 2015.